# COBias and Debias: Minimizing Language Model Pairwise Accuracy Bias via Nonlinear Integer Programming

## Abstract

When performing classification tasks with language models, would you prefer having only one highly accurate class or having every class deliver reliable performance? Obviously, a more balanced accuracy among classes better reflects the expectations of the majority of users. Especially for large language models (LLMs), the fact that they achieve a fair overall accuracy by in-context learning (ICL) obscures a large difference in individual class accuracies. In this work, we uncover and tackle language models' imbalance in per-class prediction accuracy by reconceptualizing it as the Contextual Oddity Bias (COBias), and we are the first to engage nonlinear integer programming (NIP) to debias it. Briefly, the proposed COBias metric measures accuracy differences among class pairs, with which we reveal the large per-class accuracy differences exhibited in LLMs of varied scales and families. Then we propose Debiasing as Nonlinear Integer Programming (DNIP) to correct ICL per-class probabilities towards lower COBias and higher overall accuracy. Our optimization objective is directly based on the evaluation scores by COBias and accuracy metrics, which is non-differentiable and solved by the simulated annealing metaheuristic. Evaluations on three LLMs across seven NLP classification tasks show that DNIP simultaneously achieves significant COBias reduction (-27%) and accuracy improvement (+12%) over the conventional ICL approach, suggesting that modeling pairwise class accuracy differences is a direction in pushing forward more accurate, more reliable LLM predictions.

## 1 Introduction

For large language models (LLMs), the fact that they achieve remarkable test accuracy over all class labels by few-shot in-context learning (ICL) (Brown et al., 2020) can obscure a large difference in individual class accuracies. We rethink language models' imbalance in per-class prediction accuracy and reconceptualize it as the Contextual Oddity Bias (COBias). What motivates COBias is a subtle yet overlooked observation, where a most frequently predicted class can hold the majority of wrong predictions of other classes, especially of a least frequently predicted class; it is empirically obvious that there is a difference in the most and least frequently predicted classes' accuracies. To this end, for this single pair of classes, we define COBias$_{single}$ to measure the difference in accuracy by a class $A$ compared to its "odd" class, which holds the majority wrong predictions of class $A$. Beyond these pairs that involve odd classes, the per-class accuracy difference is indeed an issue among *every* pair of classes. Extending the above measure to every pair, we propose a COBias metric which aggregates the differences in pairwise class accuracies. COBias reflects a common language model failure, seen in a variety of NLP benchmark tasks from text classification to question answering. It is a direct outcome of LLMs' liking for specific patterns or answers, leading to highly imbalanced predicted class distributions (Kassner & Schütze, 2020). This per-class accuracy difference can be further attributed to ICL prompt sensitivity or rooted in the pretraining data. To name a few, the causes encompass predilections for common tokens in the pretraining corpora (Zhao et al., 2021), choices of prompt templates (Jiang et al., 2020; Min et al., 2022; Lyu et al., 2023), demonstrations (Holtzman et al., 2021), and prompt orders (Lu et al., 2022; Turpin et al., 2023). Unintended LLM behaviors may also exacerbate the bias, such as user-following sycophancy (Perez et al., 2023; Sharma et al., 2024; Wei et al., 2024) and prompt toning effects (Jones & Steinhardt, 2022; Lin & Ng, 2023; Li

et al., 2023; Wang et al., 2024). Despite the causes, COBias renders a language model less effective for lower-accuracy classes and hinders users from trusting the answers, heightening the need for a unified debiasing method at output level.

Importantly, it is more than just mitigating a model's bias towards a class (each class), but more precisely, the model's tendency to predict other classes as it.That is why we need to model the interaction between pairwise class predictions for better bias reduction. An example of an odd class and the per-class accuracy difference are shown by the upper plots in Figure 1. Essentially, mitigating per-class accuracy bias is a critical step in pushing forward LLM inference abilities, as it directly targets improving lower-accuracy classes while improving or at least not hurting the overall accuracy.

To reduce COBias and rectify LLM outputs, we directly operate on the per-class probabilities given by the conventional ICL approach. We propose DNIP, the **D**ebiasing as **N**onlinear **I**nteger **P**rogramming method, to adjust per-class probabilities towards lower COBias and higher overall accuracy, from a combinatorial optimization viewpoint (Section 3). We formulate the debiasing problem as a discrete correction weight selection problem. In particular, DNIP re-weights per-class probability with an optimal set of weights selected based on the evaluation metrics of accuracy and COBias, and thus obtains much less biased predictions with higher accuracy.

On three widely used, different-scale LLMs across seven NLP evaluation datasets, we show the effectiveness of DNIP. Our key key messages are as follows:

- We introduce the COBias evaluation metric to assess the pairwise class accuracy bias in language model predictions, and reveal that LLMs of different scales and families consistently exhibit a large COBias score, at an average of 43%.
- We propose Debiasing as Nonlinear Integer Programming (DNIP) to jointly minimize COBias and maximize accuracy. An efficient simulated annealing algorithm is adopted to solve the mathematical model.
- Experiments show that DNIP achieves the best of both worlds for lower COBias (avg. $43\% \rightarrow 16\%$) and higher overall accuracy (avg. $52\% \rightarrow 64\%$) over the ICL approach.

## 2 THE CONTEXTUAL ODDITY BIAS AND THE DEBIASING PROBLEM

Based on the phenomenon that a class can hold the majority of wrong predictions of another class, which also empirically manifests as accuracy difference between the two classes, we define the class that the other class is most biased towards as an odd class, and their absolute accuracy difference as Contextual Oddity Bias (single), i.e., COBias$_{\text{single}}$. As the name suggests, a model's tendency to over-predict an odd answer is contextual (depending on the model or task), which may reflect surface patterns captured from the task-specific prompts or a common token in the pretraining data. For example, even on the same task, which answers are most frequently predicted can vary across different models. "CO" in COBias is also interpreted as the pairwise nature of the bias.

**The Contextual Oddity Bias Metric.** Formally, given an $N$-way classification task, let $x_m$ denote an input whose label is $y_m$. By few-shot prompting, let $\boldsymbol{p}_m = (p_{m1}, \ldots, p_{mN})$ denote the output token probability over $N$ classes $c_1, \ldots, c_N$. The ICL prediction $\hat{y}_m$ is $\arg\max_{j \in \{1,\ldots,N\}} \boldsymbol{p}_m$. For a given class $c_i$, we denote the class it is most biased towards as its oddest class $c_o$. Let $A_o$ and $A_i$ denotes the class accuracies respectively, then COBias$_{\text{single}}$ is computed by:

$$\text{COBias}_{\text{single}} = \sum_{i \neq o} \left| A_o - A_i \right| \tag{1}$$

Class accuracy $A_i$ is given by $\frac{1}{|\mathcal{C}_i|} \sum_{m \in \mathcal{C}_i} \mathbb{1}\{\hat{y}_m = y_m\}$, where $\mathcal{C}_i$ is the set of indices of class $c_i$ examples, and $\mathbb{1}(\cdot)$ returns 1 if the condition inside is satisfied and 0 otherwise.

Bedsides the oddest class $c_o$, there could be the second oddest, the third oddest, or etc. class for a given class. Moreover, the per-class accuracy difference is *the issue* to solve for enhancing LLMs' text classification abilities. Taking these into account, we generalize the above measure to every pair of classes, proposing **the COBias metric** in Equation 2:

$$\text{COBias} = \binom{N}{2}^{-1} \sum_{i=1}^{N-1} \sum_{j=i+1}^{N} \left| A_i - A_j \right| \tag{2}$$

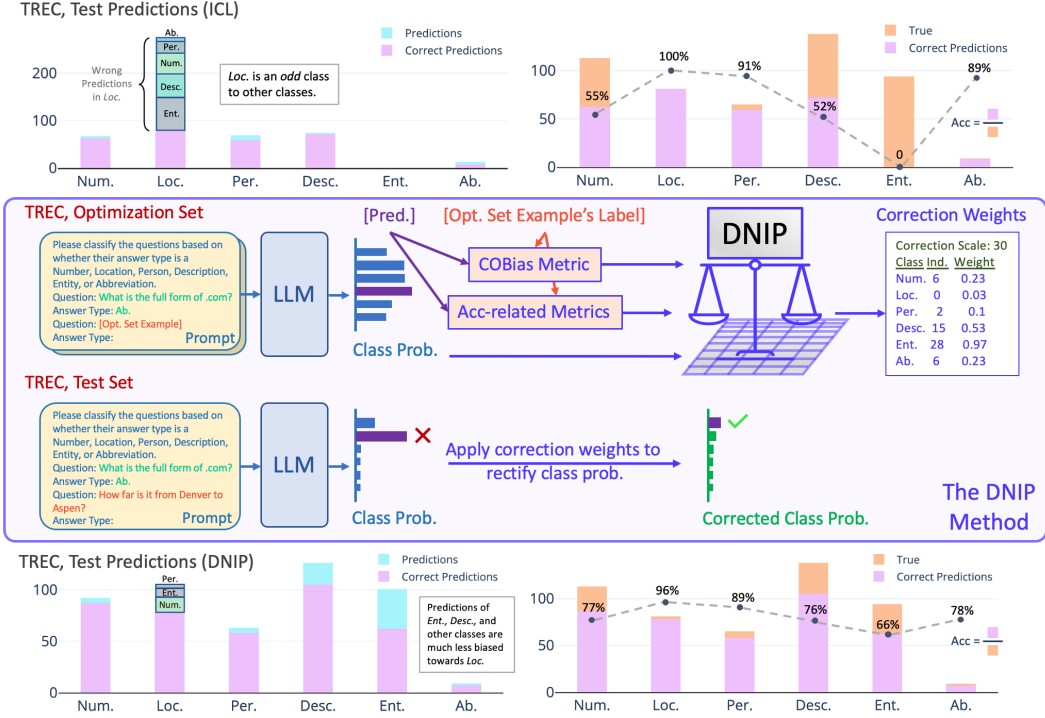

Figure 1: An overview of the DNIP method. We take the TREC (Li & Roth, 2002) task for illustration; the ICL approach uses 1-shot prompting whose prompt is given by light yellow text boxes. The upper-left plot shows that *Loc.* is an odd class to the others, holding their majority wrong predictions, especially for *Ent.*; the upper-right plot demonstrates the stark contrast in per-class accuracy, with 100% for *Loc.* and 0 for *Ent.*. The bottom plots show that DNIP greatly reduces pairwise class accuracy differences, especially boosting accuracy for *Ent.* ($0 \rightarrow 66\%$).

COBias reflects the inter-class imbalance in accuracy. Indeed, the accuracy difference of some pairs in Equation 2 does not reflect a strong biasedness between the two classes, however, it reflects a well-round assessment of both class pairs involving odd classes and those who do not.

**The Debiasing Problem.** The problem is two-fold - we need to reduce the contextual oddity bias, while maintaining or increasing the overall test accuracy.

## 3 IMPROVING LLM OUTPUTS BY *Debiasing as Nonlinear Integer Programming*

In this work, we approach the debiasing problem as a nonlinear integer programming (NIP) problem. The objective is to optimize correction weights to rectify the biased, per-class probabilities, with directly modeling the COBias metric and two accuracy-related metrics. The weights are optimized on a set of labeled examples, and are simply plugged in to inference-time probabilities. An overview of the proposed DNIP method is shown in Figure 1. Our source code will be released.

### 3.1 THE DNIP OBJECTIVE

Considering $M$ examples with known class labels, we let $\boldsymbol{p}_m$ denote the normalized output probability over $N$ class tokens for an example, and $\boldsymbol{p}_m = (p_{m1}, \ldots, p_{mn}, \ldots, p_{mN})$, $\sum_{n=1}^{N} p_{mn} = 1$, $m \in \{1, \ldots, M\}$. To debias $\boldsymbol{p}_m$, we introduce a discrete $K$-point correction scale $\boldsymbol{\omega}$ ranging from 0 to 1, to re-weight class probability. Let $\boldsymbol{\omega} = (\omega_1, \ldots, \omega_K)$ denote the parameter for the discrete weights in the correction scale, where $\omega_k = k/K$. An example 10-point scale is given by $\boldsymbol{\omega} = (0.1, \ldots, 1.0)$. For each class probability in $\boldsymbol{p}_m$, it multiplies by a correction weight chosen from $\boldsymbol{\omega}$ to obtain a debiased class probability.

Finding optimal correction weights is a combinatorial optimization problem: for $N$-way class probability $\boldsymbol{p}_m$, each class selects a weight from the $K$-dimensional correction scale. To find the optimal combination, the brute force strategy is an enumerated search, where the number of weight combinations to explore is $K^N$ for an example; for $M$ examples, it results in a computational complexity of $\mathcal{O}(K^N \cdot M)$, which can not run in polynomial time, calling for smarter methods.

Instead, we model the debiasing problem as a nonlinear integer program, and search for an optimal set of class correction weights with simulated annealing. To this end, the problem boils down to selecting optimal weight indices, and we define an integer selection variable $\xi = (\xi_1, \ldots, \xi_N)$ that selects a weight index $k$ for the $n$-th class probability in $\boldsymbol{p}_m$, namely:

$$\xi_n = k \quad \text{if the } n\text{-th class selects } \omega_k. \tag{3}$$

The DNIP objective function and constraints are formally given by:

$$\min \quad Z(\xi) = \frac{1}{M} \sum_{m=1}^{M} \mathbb{1}\{\hat{y}_m \neq y_m\} + \beta \binom{N}{2}^{-1} \sum_{i=1}^{N-1} \sum_{j=i+1}^{N} \left| A_i - A_j \right| - \tau \sum_{j=1}^{N} \mathrm{PMI}_j \tag{4}$$

$$\text{s.t.} \quad \hat{y}_m = \underset{n \in \{1,\ldots,N\}}{\arg\max} \; \omega_{\xi_n} p_{mn}, \qquad m = 1, \ldots, M \tag{5}$$

$$A_i = \frac{1}{|\mathcal{C}_i|} \sum_{m \in \mathcal{C}_i} \mathbb{1}\{\hat{y}_m = y_m\}, \qquad i = 1, \ldots, N \tag{6}$$

$$\mathrm{PMI}_j = \mathrm{PMI}(\hat{S}^j, S^j) = \log \frac{f(\hat{S}^j, S^j)}{f(\hat{S}^j) f(S^j)}, \qquad j = 1, \ldots, N \tag{7}$$

$$\xi_n \in \{1, \ldots, K\}, \qquad n = 1, \ldots, N \tag{8}$$

where $\beta$ and $\tau$ are parameters that control the bias and the negative PMI terms respectively, $\hat{S}^j$ and $S^j$ denote the examples of prediction $j$ and true label $j$ respectively[1]. The debiased class probabilities and prediction are:

$$\boldsymbol{p}_m^* = (\omega_{\xi_1} p_{m1}, \ldots, \omega_{\xi_N} p_{mN}) \tag{9}$$

$$y_m^* = \underset{n \in \{1,\ldots,N\}}{\arg\max} \; \boldsymbol{p}_m^* \tag{10}$$

The intuition is to encourage the adjusted predictions to have lower inter-class bias while improving overall accuracy. As the second term COBias has minimized the inter-class accuracy bias, the first term error rate brings the class prediction closer to the actual class, and the third term negative PMI further enforces the class prediction to be more confident when it is close to the actual class. The PMI term can also be seen as a constraint to penalize each individual class.

## 3.2 Solving DNIP with Simulated Annealing

We solve DNIP by simulated annealing (Kirkpatrick et al., 1983; Eglese, 1990). Simulated annealing (SA) is a Metropolis-Hastings sampling algorithm, which is widely adopted to solve optimization problems with its versatility in dealing with highly nonlinear models, noisy data, and many constraints (Busetti, 2001). The DNIP mathematical model is such a case, pointing to the need of SA. We especially take advantages of SA algorithms' searching ability - escaping from local optima and finding global optima after reaching local optima - to find optimal correction weights. In details, the SA strategy consists of searching the solution space starting from a randomly initialized $\xi$ and then generating a new one by perturbing it. The new solution is either accepted or rejected by the Metropolis criterion after evaluating the objectives.

---

[1]In details, $f(\hat{S}^j)$ is the ratio between the number of examples with prediction $j$ and the total number of examples, similarly, $f(S^j)$ is the ratio between the number of examples with class label $j$ and the total number of examples, $f(\hat{S}^j, S^j)$ is the ratio between the number of correct predictions of class $j$ and the total number of examples. Therefore, $\mathrm{PMI}_j = \log\left(\frac{\mathrm{n}(\hat{S}^j, S^j)/N}{\mathrm{n}(\hat{S}^j)/N \cdot \mathrm{n}(S^j)/N}\right)$, where $\mathrm{n}(\cdot)$ is the count. In actual computations, we apply add-$u$ smoothing to smooth out zeros in the numerator and denominator. The value of the smoothing coefficient $u$ is selected along other parameters on a development set.

Algorithm 1 describes the searching for optimal $\xi$ using SA. The outer loop performs the cooling/annealing process. In this work, a geometric decay with $\alpha = 0.95$ and initial temperature of 200,000 is set as cooling schedule. The inner loop simulates the thermal equilibrium reaching process at a temperature, and its criterion ensures that either the number of generated or accepted solutions at a temperature is above a threshold Romeo & Sangiovanni-Vincentelli (1991). We sample a new $\xi$ from the neighborhood by randomly substitute a selection in $\xi$. The computational complexity for the SA algorithm is $\mathcal{O}(NK)$. Please refer to Appendix A for derivations.

---

**Algorithm 1** Optimizing the selection of correction weights with simulated annealing

---

**Input:** $(\xi, \boldsymbol{p}, y, \omega)$: weight selection variable, class probabilities and labels, the weight scale
**Output:** $(\xi, \{y_m^*\}_1^M)$
 1: $T \leftarrow T_{max}, \xi^* \leftarrow \xi \leftarrow$ **INIT**()
 2: **while** $T \geq T_{min}$ **do**
 3:     **while** *inner-loop criterion* is not satisfied **do**
 4:         $\xi_{new} \leftarrow$ **PERTURB**$(\xi)$     ▷ Sample a new $\xi$
 5:         $\Delta z \leftarrow z(\xi_{new}) - z(\xi)$
 6:         **if** $\Delta z \leq 0$ **then**
 7:             $\xi \leftarrow \xi_{new}$
 8:             **if** $z(\xi_{new}) < z(\xi^*)$ **then**
 9:                 $\xi^* \leftarrow \xi_{new}$
10:         **else if** RANDOM$(0, 1) <$ EXP$(\Delta z/T)$ **then**
11:             $\xi \leftarrow \xi_{new}$     ▷ Accept a worse $\xi$
12:     $T \leftarrow \alpha T$
13: $y^* \leftarrow$ **INFER**$(\xi^*, \boldsymbol{p}, \omega)$
14: **return** $\xi^*, y^*$

---

## 4 EXPERIMENTS

### 4.1 EXPERIMENTAL SETUP

**Evaluation Tasks.** The proposed method is evaluated on a diverse set of binary and multi-class NLP classification tasks, across general domains and biomedical domains. The five general-domain evaluation datasets are 4-way news topic classification AGNews (Zhang et al., 2015), 14-way ontology classification DBpedia (Auer et al., 2007), 5-way sentiment classification SST-5 (Socher et al., 2013), 6-way retrieval question classification TREC (Voorhees & Tice, 2000; Li & Roth, 2002), binary entailment recognition RTE (Dagan et al., 2006); the two domain-specific datasets from BLURB (Gu et al., 2021) are 5-way biomedical relation extraction DDI (Segura-Bedmar et al., 2013), and 3-way biomedical question answering PubMedQA (Jin et al., 2019). The selected biomedical tasks are of practical uses. For example, detecting DDI is useful in preventing adverse effects from drug combinations.

Each dataset comes with a training and a test set. We make use of the default training set for DNIP optimization, except for AGNews, DBpedia, and DDI, for which we randomly select 10,000 training examples since the datasets are quite large. We then split the training examples into 95%, 5% optimization and development sets. We evaluate on the default test set except for AGNews, for which we evaluate on 5,000 randomly selected test examples. Evaluation metrics are COBias and accuracy.

**Models and Experimental Configuration.** We evaluate three open-source LLMs across varied scales and model families, including GPT-2-XL (1.5B parameters), Llama-2-7B (7B parameters), and Llama-2-13B (13B parameters). For ICL, we use 1-shot prompting, and obtain the output per-class softmax probabilities. To standardize calculations, we normalize the softmax probabilities over all classes. For DNIP, we tune parameters of $\beta, \tau, K, u$ on the development set. To account for variance caused by different demonstrations, for each model and dataset, we perform three runs of prompting with different 1-shot demonstrations and obtain three sets of initial probabilities. For both ICL and DNIP, we report the average test accuracy and COBias over the three runs. The demonstration is randomly selected from training examples. All prompting is done on an NVIDIA A100 GPU. The simulated annealing algorithm executes on CPU (execution time depends on the optimization set size, 3 to 30 minutes on AMD EPYC 7742 CPU or slightly longer on local laptop CPU).

### 4.2 MAIN RESULTS

The average test accuracy (in black) and COBias (in blue) with standard deviation over three runs are shown in Table 2. We find that test COBias by ICL is large in all three LLMs, with GPT-2-XL having the largest average COBias of 50.3%, showing that LLMs of different sizes and families exhibit large per-class accuracy differences. With DNIP, the average test COBias over three models reduces from

| Model | Eval. Metric | AGNews | DBpedia | SST-5 | TREC | RTE | DDI | PubMedaQA | Avg. |
|---|---|---|---|---|---|---|---|---|---|
| | | *1-shot ICL* | | | | | | | |
| GPT-2-XL | Acc | $52.1_{5.4}$ | $31.8_{9.9}$ | $34.9_{13.7}$ | $27.4_{10.5}$ | $\mathbf{55.4_{1.9}}$ | $14.5_{4.4}$ | $55.2_{0.0}$ | 38.8 |
| | COBias | $35.5_{11.5}$ | $40.0_{3.6}$ | $48.7_{5.4}$ | $45.6_{8.7}$ | $82.4_{24.5}$ | $40.7_{5.9}$ | $59.4_{12.6}$ | 50.3 |
| Llama-2-7B | Acc | $86.4_{2.5}$ | $88.9_{2.0}$ | $42.1_{11.1}$ | $66.7_{6.6}$ | $66.3_{4.3}$ | $6.7_{0.4}$ | $40.3_{6.7}$ | 56.8 |
| | COBias | $14.0_{6.5}$ | $13.5_{2.1}$ | $55.6_{1.5}$ | $33.2_{10.0}$ | $61.6_{10.5}$ | $41.4_{1.7}$ | $40.9_{16.1}$ | 37.2 |
| Llama-2-13B | Acc | $79.9_{7.0}$ | $88.6_{1.7}$ | $44.9_{4.3}$ | $68.5_{10.8}$ | $71.5_{2.2}$ | $7.2_{0.9}$ | $55.1_{2.9}$ | 59.4 |
| | COBias | $28.3_{16.1}$ | $16.2_{3.7}$ | $53.1_{5.0}$ | $35.9_{6.5}$ | $43.4_{7.0}$ | $45.6_{5.9}$ | $61.2_{1.9}$ | 40.5 |
| | | *DNIP* | | | | | | | |
| GPT-2-XL | Acc | $\mathbf{68.5_{1.0}}$ | $\mathbf{69.9_{9.1}}$ | $\mathbf{44.5_{2.20}}$ | $\mathbf{46.3_{12.7}}$ | $50.8_{2.1}$ | $\mathbf{43.9_{14.9}}$ | $\mathbf{57.1_{1.3}}$ | $\mathbf{54.4}$ |
| | COBias | $\mathbf{1.4_{0.5}}$ | $\mathbf{24.1_{8.3}}$ | $\mathbf{26.0_{2.5}}$ | $\mathbf{27.2_{7.2}}$ | $\mathbf{7.1_{5.0}}$ | $\mathbf{17.0_{7.1}}$ | $\mathbf{29.8_{25.0}}$ | $\mathbf{18.9}$ |
| Llama-2-7B | Acc | $\mathbf{86.7_{0.4}}$ | $\mathbf{92.9_{0.4}}$ | $\mathbf{50.6_{2.7}}$ | $\mathbf{68.1_{1.0}}$ | $\mathbf{73.9_{2.3}}$ | $\mathbf{44.5_{3.8}}$ | $\mathbf{62.7_{8.3}}$ | $\mathbf{68.5}$ |
| | COBias | $\mathbf{1.3_{0.1}}$ | $\mathbf{7.7_{0.6}}$ | $\mathbf{28.0_{21.6}}$ | $\mathbf{15.9_{1.6}}$ | $\mathbf{1.9_{1.8}}$ | $\mathbf{11.6_{3.0}}$ | $\mathbf{35.4_{22.8}}$ | $\mathbf{14.5}$ |
| Llama-2-13B | Acc | $\mathbf{87.9_{0.7}}$ | $\mathbf{93.4_{0.6}}$ | $\mathbf{48.3_{1.9}}$ | $\mathbf{77.1_{2.0}}$ | $\mathbf{74.3_{0.8}}$ | $\mathbf{40.4_{6.0}}$ | $\mathbf{63.1_{14.0}}$ | $\mathbf{69.2}$ |
| | COBias | $\mathbf{6.3_{0.6}}$ | $\mathbf{7.7_{0.6}}$ | $\mathbf{18.7_{10.1}}$ | $\mathbf{14.2_{1.3}}$ | $\mathbf{4.3_{3.3}}$ | $\mathbf{7.5_{3.2}}$ | $\mathbf{41.1_{29.6}}$ | $\mathbf{14.3}$ |

Table 1: Performance comparisons between the 1-shot ICL approach and DNIP. In each cell, we report the average test accuracy/COBias (%) and standard deviation over three different prompts. DNIP greatly improves test accuracy and significantly lowers test COBias upon the ICL approach.

43% to 16%, and the test accuracy increases from 52% to 64%. Specifically, DNIP achieves an 16%, 12%, and 10% absolute improvement in test accuracy for GPT-2-XL, Llama-2-7B, and Llama-2-13B respectively; meanwhile, DNIP significantly reduces test COBias by an absolute 29%, 23%, and 26% for the three models, over ICL results. As for detailed results, for GPT-2-XL where the ICL test accuracy is below 30% (DDI), DNIP boosts test accuracy by 29% while lowering test COBias from 41% to 17%; for Llama-2 7B and 13B where the ICL test accuracy is already in higher 80s, e.g., DBpedia, DNIP can further improve the accuracy to 90s while reducing COBias. In addition, we observe a very large ICL test COBias at 82.4% for GPT-2-XL on RTE. After applying DNIP, the COBias greatly reduces to 7.1%. The 82.4% COBias on RTE suggests that the conventional ICL predictions mostly fall in a single class, failing to predict the other class. In fact, The RTE task seems too challenging that GPT-2-XL's output probability of class *True* is much higher than that of class *False*. Though re-weighting can lower both probabilities, it may not gain a corrected prediction. Still, the over 75% absolute reduction in COBias demonstrates that DNIP in the worst case can balance severely skewed ICL predictions without hurting the overall accuracy much. On two biomedical tasks, DNIP brings an average absolute increase of 34% and 11% in accuracy, and an average absolute reduction of 31% and 13% in COBias, showing its effectiveness for domain-specific tasks.

These results show that DNIP strongly mitigates COBias and greatly improves the accuracy of lower-performing classes, while boosting overall accuracy. Moreover, relatively large LLMs can suffer even higher COBias than smaller models on biomedical tasks, suggesting that the per-class accuracy bias does not go away as models scale, and debiasing is essentially needed to improve inference for larger LLMs. Furthermore, on all datasets, the computational time is in the scale of minutes, ranging from a few minutes to dozens of minutes, even for tasks of more than 10 classes. Last but not least, for most of the datasets experimented in this paper, a weight scale of 30 is good enough, suggesting the effectiveness of DNIP without requiring a highly fine-grained weight scale.

### 4.3 DNIP BALANCES CLASS ACCURACIES ACROSS VARIED CLASSIFICATION TASKS, AS MANY AS 14 CLASSES

We visualize pairwise class accuracy differences before and after applying DNIP on Llama-2-13B by the heatmaps in Figure 2, which show that DNIP is straightforwardly effective in reducing COBias. In each heatmap, the value of row $i$, column $j$ cell represents the absolute difference between the test accuracies of two classes $i, j$, i.e., $|A_i - A_j|$. We observe that DNIP brings the accuracy differences down to close to zero for 4 out of 7 datasets, including AGNews, DDI, DBpedia, and RTE; and the most relative reduction is seen on TREC. These results show that DNIP consistently reduces prediction bias across different NLP classification tasks.

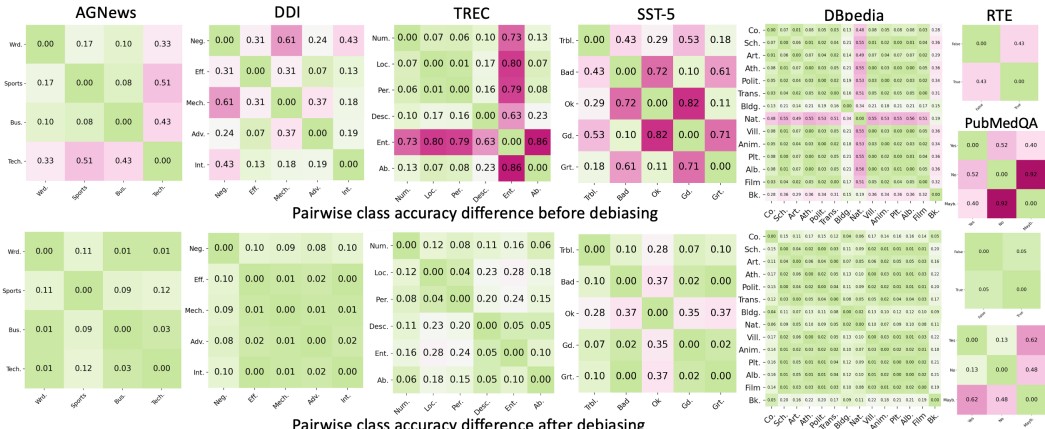

Figure 2: Comparisons of pairwise class accuracy differences on test sets before and after applying DNIP; the pinker the higher difference, the greener the lower difference; average accuracy over three runs are used. Heatmaps after debiasing show nearly 0 differences for many class pairs, demonstrating DNIP's effectiveness for COBias reduction.

**Relieving Odd Classes for 14-Way Ontology Classification.** We further investigate the 14-way classification task DBpedia. Before debiasing, the *Nat.* class exhibits the largest class accuracy difference compared to other classes, suggesting that *Nat.* is either the highest-accuracy class manifesting a strong prediction bias towards *Nat.*, or the lowest-accuracy class where predictions of true *Nat.* examples are biased to one or more odd classes. We plot the confusion matrix of test results of 1 run out of 3 runs in Figure 3 and find that *Nat.* has the lowest accuracy, and its predictions are biased to two odd classes, *Trans.* and *Bldg.*, with *Trans.* being the oddest to *Nat.*. DNIP greatly reduces false negatives for *Nat.*, i.e., reduces wrong *Trans.* and *Bldg.* predictions for *Nat.*. Besides these visualizations, on an average over three runs, the test accuracy for *Nat.* improves from 44% to 89%, and that for *Trans.* also increases from 78% to 87%, demonstrating DNIP's effectiveness in reducing pairwise class accuracy bias for classification tasks of as many as 14 classes.

Furthermore, we see more balanced class accuracies with DNIP for smaller LLMs of Llama-2-7B and GPT-2-XL, in Appendix B.

### 4.4 DNIP OBJECTIVE ABLATION

We present ablation analysis on objective functions to show that the integration of accuracy-related objectives and the COBias objective are indispensable in achieving the best of both worlds. Recall three parts in our objective function $z$: the error rate term $z_1 = \frac{1}{M} \sum_{m=1}^{M} \mathbb{1}\{\hat{y}_m \neq y_m\}$; the

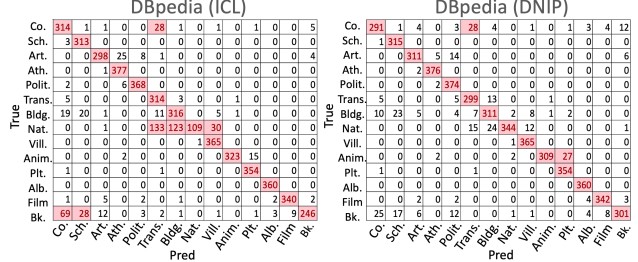

Figure 3: Test confusion matrix for DBpedia before and after applying DNIP. DNIP greatly reduces wrong predictions for *Nat.*, which is the lowest-accuracy class before debiasing.

COBias term $z_2 = \binom{N}{2}^{-1} \sum_{i=1}^{N-1} \sum_{j=i+1}^{N} |A_i - A_j|$; and the PMI term $z_3 = \sum_{j=1}^{N} \text{PMI}_j$; the accuracy-related terms are $z_1$ and $z_3$. We set up 7 different objective functions to perform DNIP: $z_1, z_2, z_3, z_1 + \beta z_2, z_1 - \tau z_3, \beta z_2 - \tau z_3, z_1 + \beta z_2 - \tau z_3$, and compare their effectiveness on Llama-2-13B in Figure 4. Ablations on GPT-2-XL and Llama-2-7B show similar findings (Appendix C). To make fair comparisons, we fix $\beta = 2.7, \tau = 0.2$ and weight scale $K = 30$, which may not result in the same scores reported in Table 2.

Among all ablations, $z_1 + \beta z_2 - \tau z_3$ achieves a balance point between accuracy and COBias. For the criteria, lower COBias is our top priority; when COBias scores are similarly low, we prefer the one that achieves higher accuracy. Compared to $z_1 + \beta z_2 - \tau z_3 z_1$: (1) Using $z_1$ or $z_3$ solely emphasizes

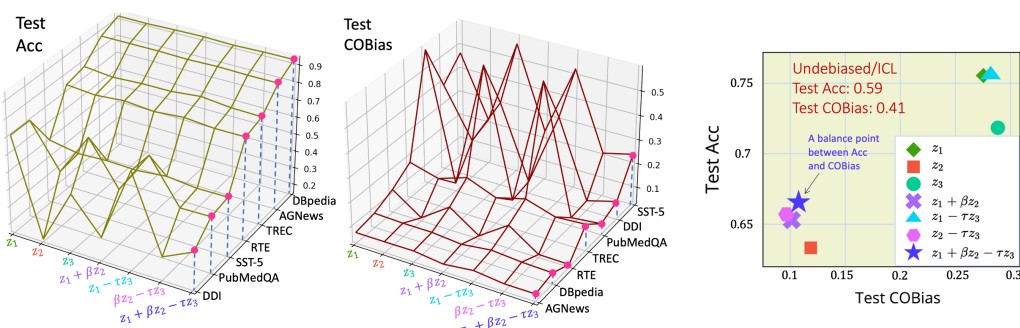

With $z_1 + \beta z_2 - \tau z_3$, COBias reduces by 30% and Acc increases by 8%, and all datasets achieve a balance between Acc and COBias, where AGNews, DBpedia each obtains its lowest COBias and highest Acc, RTE obtains its lowest COBias and second highest Acc, and the rest of the four datasets obtain a low COBias with a fair Acc. Notably, the accuracy-related objectives alone can raise Acc from 59% to over 75%, but COBias reduction is 13%.

Figure 4: Ablation analysis of the objective function, exemplified with Llama-2-13B; test scores averaged over 7 datasets are used for the rightmost plot.

improving accuracy; though obtaining higher or similar accuracy, they suffer a higher COBias. (2) Using $z_1 - \tau z_3$ also obtains a higher/similar accuracy but a higher COBias. (3) Using $z_2$ solely focuses on mitigating COBias and is straightforwardly effective in reducing COBias. However, the shortcoming is obvious: a lower accuracy. (4) Using $z_1 + \beta z_2$ or $\beta z_2 - \tau z_3$ is promising with a similar/lower accuracy and a slightly lower COBias. We note that accuracy-related objectives alone increase accuracy by over 16% ($75\% - 59\%$), which is greater than using the objective combining COBias, suggesting the NIP can be a direction for those who look for accuracy improvement.

### 4.5 PERFORMANCE IMPROVEMENTS VS. NUMBER OF OPTIMIZATION EXAMPLES

We evaluate DNIP with varying optimization examples on Llama-2-13B. We find that DNIP requires only a small optimization set of 10 examples for a fair improvement in both COBias and accuracy, and it exhibits a further, emergent COBias reduction at several thousands optimization examples.

**Test COBias reduction reaches 9% to 72% at 10 optimization examples, and continues to grow with more optimization examples.** Figure 5 shows the relative test COBias reduction over ICL results. At 10 examples, DNIP can achieve a relative COBias reduction ranging from 9% to 72%. We observe an overall trend of COBias reduction except for DDI and PubMedQA, who have a spike at 100 examples, but DDI rises again after 500 examples while PubMedQA levels off (PubMedQA full optimization size: 950). An explanation is that DNIP prioritizes COBias reduction more than accuracy increase at fewer examples; at more examples, the accuracy-related objectives show more influence. It also indicates that domain-specific datasets are more sensitive to the trade-off. We further notice a surge in reductions from 1,000 to full for DDI, SST-5, and

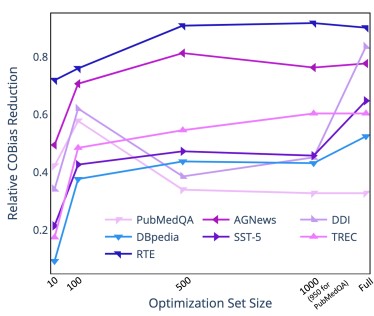

Figure 5: Relative test COBias reduction; higher is better.

DBpedia (full optimization sizes: 9,500, 8,116, 9,500), suggesting that around 8,000 examples are needed for a further emergent reduction - PubMedQA may see further COBias reduction if more optimization examples are available.

**Test accuracy is improved with only 10 optimization examples, and plateaus at around 500 examples.** Figure 6 shows that, at 10 examples, the relative increase in accuracy ranges from 0.2% to 11% except DDI, which is 160%. At 100 examples, the increase ranges from 3% to 208% (DDI, $7.2\% \rightarrow 18.7\%$). The increase continues to grow and stabilizes at around 500 examples. At 1,000 examples, test accuracy for DDI rises to 52%, which is more than 6 times of the ICL accuracy, showing that DNIP is effective for this challenging biomedical NLP task. Moreover, we observe for RTE a declining trend of accuracy increase, which matches its growing trend of COBias reduction, reflecting the trade-off between the two objectives. We also note a slight decrease in accuracy improvement for

SST-5 and DDI at full size, which matches their sudden increase in COBias reduction, verifying that DNIP has emergent COBias reduction abilities with thousands of optimization examples.

Overall, DNIP improves both accuracy and CO-Bias at 10 examples, and achieves acceptable CO-Bias reduction and accuracy improvement at 100 examples, with test COBias reduced by 57% and test accuracy improved. For detailed scores, please refer to Appendix D.

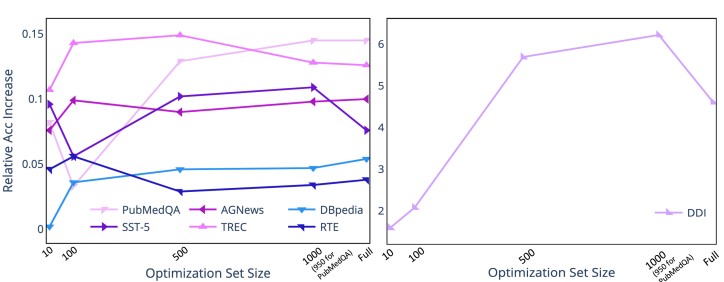

Figure 6: Relative test accuracy increase; higher is better.

## 5 RELATED WORK

LLMs may fail in various ways. Analysis towards understanding LLM failure modes has gained remarkable insights on how they fail and how to identify/avoid failures. For example, understanding hallucinations in LLMs facilitates fact-checking evaluation metrics for LLMs.Ji et al. (2023); Dhingra et al. (2019); Guo et al. (2022). Jones & Steinhardt (2022) reveals failure modes of LLMs which resemble cognitive biases and measures the failures on GPT-3.5 and Codex. Lu et al. (2022) identify ICL order sensitivity that result in LLM prompting failures. Turpin et al. (2023) shows that LLM reasoning abilities are affected by adding biasing features to the input. Along this line, we tackle a common LLM failure of the per-class accuracy difference, via nonlinear integer programming optimization on the evaluation metric scores.

## 6 DISCUSSION

**More Methods That Potentially Improve Accuracy while Reducing COBias.** Prior methods tackle the prediction bias problem from different angles. Although COBias is not be explicitly modeled, they are promising to reduce COBias. Therefore, we draw comparisons to two popular categories of debiasing methods, adaptations and calibrations.

In particular, with the same class probability vectors as input, we build a single linear adapter (LA) network with adapter dimension of 64, and train it towards the same objective as

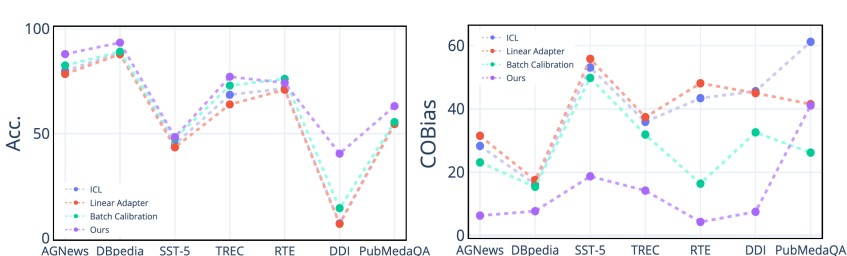

Figure 7: Comparisons to more methods.

DNIP; we also follow the batch calibration method (Zhou et al., 2024) to estimate a calibration term using test examples. Figure 7 shows the comparison results. As can be seen, finetuning a simple linear adapter does not work well on the class output probabilities. The proposed method outperforms both methods in accuracy and COBias.

**More ICL Settings.** To see how DNIP scales with more diverse prompts, we prompted Llama-2-13B in two additional settings: 5-shot, where each demonstration is randomly selected as in the 1-shot case; k-shot, where k is the number of classes and each class is represented by a demonstration example in the prompt. Results are shown in Appendix E. DNIP significantly reduces COBias and improves accuracy in both settings, further showcasing the effectiveness of our approach.

**Using Chain-of-Thought Prompting to Improve LLM Predictions.** Some may argue that a simple remedy for biased predictions is Chain-of-Thought (CoT) prompting, which elicit more reasoning in

LLMs. However, Turpin et al. (2023) points out that CoT explanations can be unfaithful, as they can be biased towards class A when prompts always put letter A as the answer, causing 36% accuracy drops on BBH (Suzgun et al., 2023). We add that prompt-based models may fall for pattern matching instead of recalling learned factual knowledge (Kassner & Schütze, 2020). Therefore, CoT may not help due to unfaithful explanations.

**Using Calibration Methods to Mitigate ICL Biases.** Previous calibration methods reduce biases caused by prompting. For example, Zhao et al. (2021); Zhou et al. (2024) use content-free or content-based prompts to measure an offset and calibrate class probabilities by removing it. This may be good for mitigating bias towards a single class, but it does not consider the interaction between class predictions, e.g., pairwise class accuracy bias studied in this work. Another hazard is using prompts to compute the offset, which may cascade biases brought by a certain or several prompt templates, and biases inherited from pretraining may not be reflected. Instead, we correcct biased predictions without more prompting, but we differ in that our correction weights are explicitly optimized on COBias and accuracy metrics, while the cluster-based decision boundary is implicitly learned.

**Evaluating Closed LLMs.** While closed LLMs such as ChatGPT and GPT-4 (OpenAI et al., 2024) are interesting to evaluate, they do not return logits/softmax probabilities over the entire vocabulary at an output token. For example, ChatGPT returns log probabilities of up to 20 most likely output tokens, and may not cover the probability for every class that is needed in our class prediction debiasing.

**Optimization Algorithms for Integer Programming.** The classic solvers for integer programming include operational research techniques such as Branch and Bound (BnB), commonly used for linear integer programming problems. However, it is difficult for these methods to solve our DNIP model, which contains discontinuous functions and is non-differentiable. In this case, a series of metaheuristic algorithms are more suitable, solversincluding Simulated Annealing (SA), Genetic Algorithms, Ant Colony Optimization, and Particle Swarm Optimization. These algorithms all belong to the same category, and each meta heuristic has their own pros and cons. Therefore, SA in this paper has a clear purpose of tackling our NIP mathematical model that classical operational research methods cannot solve.

**Our Motivation Is Different from Classical Post-hoc Corrections.** Some may contend that achieving equitable accuracies across all classes is a well-explored issue in standard machine learning classifiers. However, it is crucial to see that per-class prediction accuracy imbalances must be understood within their specific context. The accuracy bias in LLMs' outputs arises from fundamentally different causes than the unequal class accuracies seen in potentially overfitted traditional classifiers. In the former, biases stem from prompts and pre-training, while in the latter, the imbalance is driven by skewness in the supervised training data. This distinction underscores why our approach is uniquely applied to LLMs' output token class probabilities.

## 7 CONCLUSION

This work tackles the imbalance in per-class prediction accuracy of language models. We introduce COBias and reveal that LLMs of varied scales and families exhibit large COBias at inference. To minimize COBias and push for higher accuracy, we view debiasing as a combinatorial optimization problem and propose DNIP, which optimizes class correction weights by jointly minimizing evaluation scores by COBias and maximizing accuracy. Empirical evaluations on three LLMs across seven multi-class NLP classification tasks demonstrate that DNIP simultaneously achieves lower test COBias and higher test accuracy over conventional ICL. We advocate that reducing COBias is a direction to gain more accurate, reliable LLM predictions. For future works, we may extend our studies to more tasks and other modalities. We aim to find simple inference-time plug-in solution for further boosting language model prediction performances.

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

# A    DERIVATION OF SA ALGORITHM COMPLEXITY

Given $N$ classes and $K$ scales, the DNIP objective has $NK$ variables. For each inner loop of SA, the length of the Markov chain is often several times of the size of variables, i.e., $\lambda NK$. For a geometric cooling schedule $T = \alpha^n T_{\max}$, the number of outer-loop iterations is:

$$N^{\text{outer}} = \log_\alpha \frac{T_{\min}}{T_{\max}} \tag{11}$$

The sampling times for the SA algorithm is $\lambda NK \log_\alpha (T_{\min}/T_{\max})$. Therefore, the computational complexity is $\mathcal{O}(\lambda NK \log_\alpha (T_{\min}/T_{\max})) = \mathcal{O}(NK)$.

# B    ADDITIONAL HEATMAPS FOR PAIRWISE ACCURACY DIFFERENCES

Figure 8 and 9 demonstrate that more balanced class accuracies are achieved with DNIP for smaller LLMs of Llama-2-7B and GPT-2-XL.

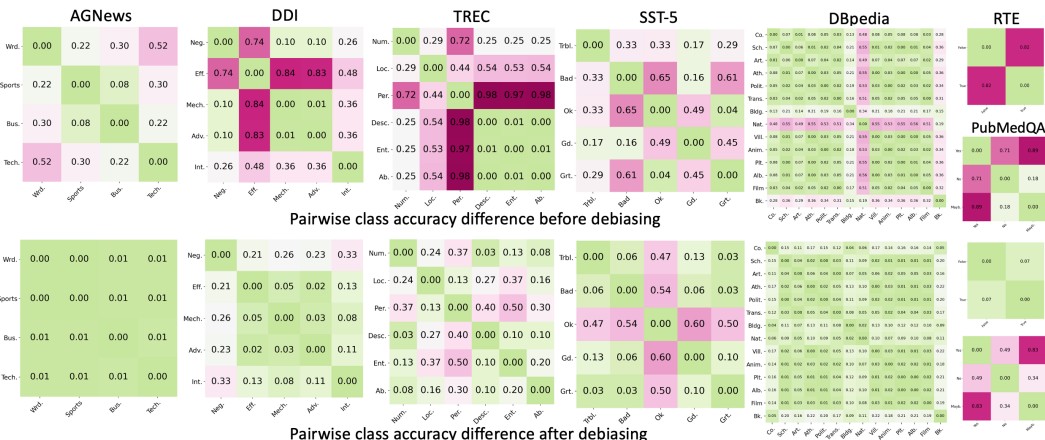

Figure 8: Comparisons of pairwise test class accuracy differences before and after applying DNIP on GPT-2-XL; the pinker the higher accuracy difference, the greener the lower accuracy difference; average accuracy over three runs are used.

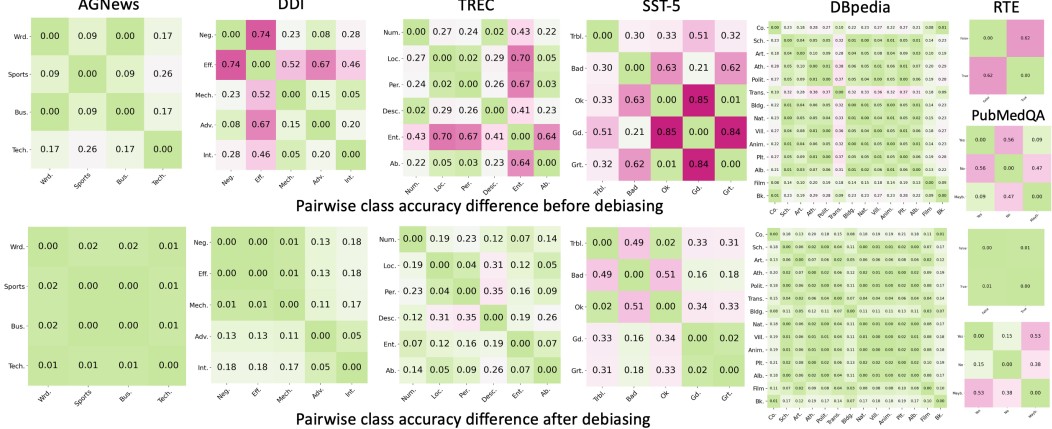

Figure 9: Comparisons of pairwise test class accuracy differences before and after applying DNIP on Llama-2-7B; the pinker the higher accuracy difference, the greener the lower accuracy difference; average accuracy over three runs are used.

## C ADDITIONAL ABLATION ANALYSIS

We show ablation analysis on GPT-2-XL and Llama-2-7B in Figure 10 and 11.

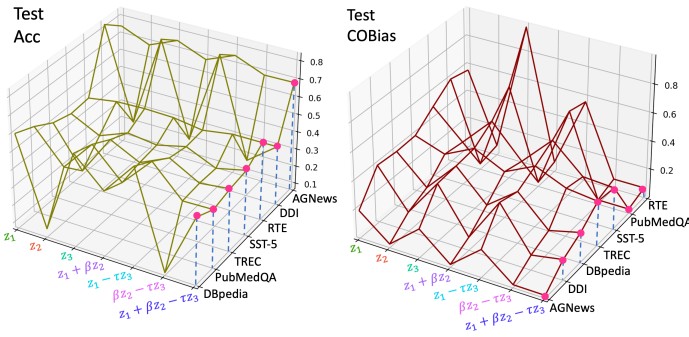

Figure 10: Ablations on objective function, GPT-2-XL, demonstrating that $z_1 + \beta z_2 - \tau z_3$ achieves a balance point between accuracy and COBias.

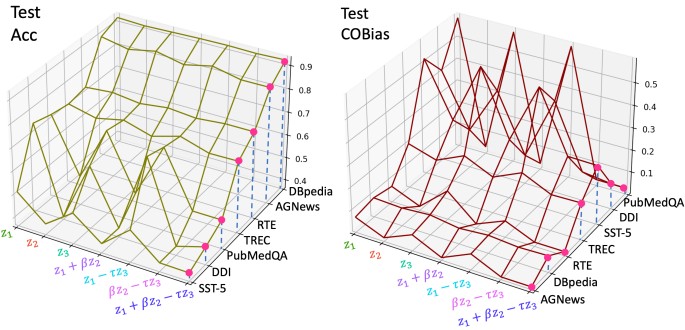

Figure 11: Ablations on objective function, Llama-2-7B, demonstrating that $z_1 + \beta z_2 - \tau z_3$ achieves a balance point between accuracy and COBias.

| Prompting Method | Metric | AGNews | DBpedia | SST-5 | TREC | RTE | DDI | PubMedQA | Avg. |
|---|---|---|---|---|---|---|---|---|---|
| 5-shot ICL | Acc | $82.5_{2.0}$ | $93.6_{1.3}$ | $45.8_{6.0}$ | $58.7_{23.3}$ | $61.9_{16.9}$ | $34.1_{42.1}$ | $44.7_{5.7}$ | $60.2$ |
| | COBias | $16.5_{6.0}$ | $9.0_{2.0}$ | $48.0_{15.4}$ | $35.4_{13.7}$ | $61.3_{40.4}$ | $44.4_{5.4}$ | $52.2_{19.3}$ | $38.1$ |
| 5-shot DNIP | Acc | $\mathbf{88.5_{0.6}}$ | $\mathbf{95.8_{0.7}}$ | $\mathbf{52.9_{2.5}}$ | $\mathbf{76.6_{8.7}}$ | $\mathbf{75.8_{4.5}}$ | $\mathbf{54.1_{5.1}}$ | $\mathbf{59.9_{1.4}}$ | $\mathbf{71.9}$ |
| | COBias | $\mathbf{7.0_{0.9}}$ | $\mathbf{5.7_{1.1}}$ | $\mathbf{15.8_{12.6}}$ | $\mathbf{14.4_{7.5}}$ | $\mathbf{3.1_{1.9}}$ | $\mathbf{16.5_{7.6}}$ | $\mathbf{9.6_{4.6}}$ | $\mathbf{10.3}$ |
| k-shot (1 demon. from each class) ICL | Acc | $83.5_{1.5}$ | $95.2_{1.2}$ | $50.3_{2.3}$ | $67.0_{12.7}$ | $75.0_{0.8}$ | $9.7_{1.0}$ | $52.3_{5.3}$ | $61.9$ |
| | COBias | $14.9_{5.1}$ | $7.0_{2.2}$ | $36.3_{7.2}$ | $38.2_{5.1}$ | $22.5_{13.2}$ | $39.7_{3.5}$ | $20.9_{4.2}$ | $25.6$ |
| k-shot (1 demon. from each class) DNIP | Acc | $\mathbf{88.7_{0.5}}$ | $\mathbf{96.6_{0.5}}$ | $\mathbf{51.3_{1.0}}$ | $\mathbf{82.7_{1.4}}$ | $\mathbf{76.7_{3.7}}$ | $\mathbf{43.6_{6.1}}$ | $\mathbf{55.3_{2.6}}$ | $\mathbf{70.7}$ |
| | COBias | $\mathbf{7.3_{0.5}}$ | $\mathbf{4.3_{0.7}}$ | $\mathbf{2.8_{1.5}}$ | $\mathbf{12.1_{5.5}}$ | $\mathbf{5.0_{3.3}}$ | $\mathbf{12.5_{4.3}}$ | $\mathbf{8.7_{1.2}}$ | $\mathbf{7.5}$ |

Table 2: DNIP results int more ICL settings. Average score over three runs are reported.

## D FULL TABLE WITH VARYING NUMBER OF OPTIMIZATION EXAMPLES FOR DNIP

The full results for all three models across varying optimization set sizes are shown in Table 3.

## E MORE ICL SETUPS

We prompted Llama-2-13B in two additional settings: 5-shot, where each demonstration is randomly selected as in the 1-shot case; k-shot, where k is the number of classes and each class is represented by a demonstration example in the prompt. Table **??** presents results on seven benchmark datasets in both settings. DNIP significantly reduces COBias and improves accuracy in both 5-shot and k-shot

| Opt. set | Eval. Metric | AGNews | DBpedia | SST-5 | TREC | RTE | DDI | PubMedaQA |
|---|---|---|---|---|---|---|---|---|
| | | | | *GPT-2-XL* | | | | |
| 0 | Acc | $52.1_{5.4}$ | $31.8_{9.9}$ | $34.9_{13.7}$ | $27.4_{10.5}$ | $\mathbf{55.4}_{1.9}$ | $14.5_{4.4}$ | $55.2_{0.0}$ |
| | COBias | $35.5_{11.5}$ | $40.0_{3.6}$ | $48.7_{5.4}$ | $45.6_{8.7}$ | $82.4_{24.5}$ | $40.7_{5.9}$ | $59.4_{12.6}$ |
| 10* | Acc | $\mathbf{72.8}_{2.1}$ | $40.6_{5.2}$ | $40.3_{3.4}$ | $\mathbf{51.5}_{5.7}$ | $50.3_{0.8}$ | $23.5_{1.0}$ | $53.7_{0.6}$ |
| | COBias | $17.2_{1.7}$ | $28.0_{6.6}$ | $40.1_{1.8}$ | $39.7_{1.0}$ | $6.5_{7.6}$ | $40.6_{3.5}$ | $41.8_{0.0}$ |
| 100 | Acc | $70.0_{5.0}$ | $41.4_{29.9}$ | $44.3_{1.3}$ | $49.0_{10.1}$ | $49.8_{2.5}$ | $42.2_{5.7}$ | $46.7_{10.8}$ |
| | COBias | $14.8_{1.3}$ | $27.8_{8.9}$ | $30.8_{3.4}$ | $31.1_{5.9}$ | $5.3_{4.9}$ | $17.1_{7.1}$ | $29.2_{16.6}$ |
| 500 | Acc | $68.2_{1.2}$ | $44.5_{33.2}$ | $43.4_{1.5}$ | $49.9_{8.5}$ | $50.5_{2.4}$ | $45.6_{9.2}$ | $49.9_{11.4}$ |
| | COBias | $2.7_{1.8}$ | $25.7_{9.6}$ | $29.8_{3.7}$ | $31.4_{6.2}$ | $\mathbf{5.2}_{3.9}$ | $24.5_{14.3}$ | $17.7_{15.8}$ |
| 1,000 | Acc | $69.6_{1.0}$ | $42.0_{32.2}$ | $42.8_{2.3}$ | $50.9_{9.0}$ | $50.3_{2.2}$ | $48.3_{4.9}$ | same as |
| | COBias | $3.0_{1.2}$ | $27.6_{11.0}$ | $28.2_{5.9}$ | $\mathbf{24.5}_{5.4}$ | $\mathbf{5.2}_{3.3}$ | $22.9_{2.5}$ | full |
| Full | Acc | $68.5_{1.0}$ | $\mathbf{69.9}_{9.1}$ | $\mathbf{44.5}_{2.20}$ | $46.3_{12.7}$ | $50.8_{2.1}$ | $43.9_{14.9}$ | $\mathbf{57.1}_{1.3}$ |
| | COBias | $\mathbf{1.4}_{0.5}$ | $\mathbf{24.1}_{8.3}$ | $\mathbf{26.0}_{2.5}$ | $27.2_{7.2}$ | $7.1_{5.0}$ | $\mathbf{17.0}_{7.1}$ | $29.8_{25.0}$ |
| | | | | *Llama-2-7B* | | | | |
| 0 | Acc | $86.4_{2.5}$ | $88.9_{2.0}$ | $42.1_{11.1}$ | $66.7_{6.6}$ | $66.3_{4.3}$ | $6.7_{0.4}$ | $40.3_{6.7}$ |
| | COBias | $14.0_{6.5}$ | $13.5_{2.1}$ | $55.6_{1.5}$ | $33.2_{10.0}$ | $61.6_{10.5}$ | $41.4_{1.7}$ | $40.9_{16.1}$ |
| 10* | Acc | $86.4_{2.5}$ | $89.9_{1.4}$ | $\mathbf{51.4}_{0.9}$ | $70.1_{0.9}$ | $\mathbf{74.9}_{2.1}$ | $31.7_{21.0}$ | $44.5_{0.6}$ |
| | COBias | $14.0_{6.5}$ | $12.5_{2.1}$ | $36.2_{3.6}$ | $22.4_{5.4}$ | $4.8_{5.1}$ | $26.7_{8.2}$ | $\mathbf{28.7}_{3.9}$ |
| 100 | Acc | $\mathbf{88.4}_{0.4}$ | $91.8_{0.7}$ | $50.9_{1.5}$ | $70.1_{1.0}$ | $73.6_{2.6}$ | $44.9_{2.5}$ | $62.6_{4.5}$ |
| | COBias | $5.8_{0.6}$ | $9.7_{1.0}$ | $34.3_{12.7}$ | $16.7_{2.4}$ | $2.9_{0.7}$ | $21.0_{6.6}$ | $35.4_{18.1}$ |
| 500 | Acc | $86.8_{0.9}$ | $92.1_{0.6}$ | $50.8_{1.7}$ | $69.7_{1.2}$ | $74.3_{2.9}$ | $\mathbf{69.3}_{1.7}$ | $\mathbf{63.5}_{7.5}$ |
| | COBias | $2.8_{0.3}$ | $8.6_{1.6}$ | $35.8_{12.3}$ | $\mathbf{15.2}_{2.1}$ | $3.1_{1.4}$ | $34.5_{0.3}$ | $37.3_{20.4}$ |
| 1,000 | Acc | $86.8_{0.3}$ | $92.5_{0.2}$ | $51.0_{2.1}$ | $69.5_{0.8}$ | $74.0_{2.5}$ | $56.5_{7.9}$ | same as |
| | COBias | $1.9_{0.5}$ | $8.0_{0.4}$ | $30.9_{20.4}$ | $15.7_{1.2}$ | $2.6_{1.2}$ | $29.5_{3.4}$ | full |
| Full | Acc | $86.7_{0.4}$ | $\mathbf{92.9}_{0.4}$ | $50.6_{2.7}$ | $68.1_{1.0}$ | $73.9_{2.3}$ | $44.5_{3.8}$ | $62.7_{8.3}$ |
| | COBias | $\mathbf{1.3}_{0.1}$ | $\mathbf{7.7}_{0.6}$ | $\mathbf{28.0}_{21.6}$ | $15.9_{1.6}$ | $\mathbf{1.9}_{1.8}$ | $\mathbf{11.6}_{3.0}$ | $35.4_{22.8}$ |
| | | | | *Llama-2-13B* | | | | |
| 0 | Acc | $79.9_{7.0}$ | $88.6_{1.7}$ | $44.9_{4.3}$ | $68.5_{10.8}$ | $71.5_{2.2}$ | $7.2_{0.9}$ | $55.1_{2.9}$ |
| | COBias | $28.3_{16.1}$ | $16.2_{3.7}$ | $53.1_{5.0}$ | $35.9_{6.5}$ | $43.4_{7.0}$ | $45.6_{5.9}$ | $61.2_{1.9}$ |
| 10* | Acc | $86.0_{1.9}/$ | $88.8_{1.6}/$ | $49.2_{0.8}/$ | $75.8_{3.2}$ | $74.8_{2.3}$ | $18.7_{12.6}$ | $59.6_{6.2}$ |
| | COBias | $14.3_{3.5}$ | $14.7_{2.5}$ | $41.7_{8.1}$ | $29.6_{4.5}$ | $12.2_{5.7}$ | $30.0_{7.8}$ | $35.3_{7.8}$ |
| 100 | Acc | $87.8_{0.2}/$ | $91.8_{0.2}/$ | $47.4_{2.0}/$ | $78.3_{2.8}$ | $\mathbf{75.5}_{1.0}$ | $22.2_{2.0}$ | $56.9_{10.4}$ |
| | COBias | $8.3_{0.5}$ | $10.1_{0.7}$ | $30.4_{4.4}$ | $18.5_{2.6}$ | $10.4_{6.9}$ | $17.3_{4.4}$ | $25.8_{16.3}$ |
| 500 | Acc | $87.1_{1.5}$ | $92.7_{0.0}/$ | $49.5_{0.7}/$ | $\mathbf{78.7}_{2.1}$ | $73.6_{1.7}$ | $48.2_{27.4}$ | $62.2_{15.8}$ |
| | COBias | $\mathbf{5.3}_{3.1}$ | $9.1_{0.3}$ | $28.0_{3.2}$ | $16.3_{1.0}$ | $4.0_{3.4}$ | $28.0_{11.5}$ | $40.4_{29.4}$ |
| 1,000 | Acc | $87.7_{0.7}/$ | $92.8_{0.1}/$ | $\mathbf{49.8}_{0.7}/$ | $77.3_{2.0}$ | $73.9_{1.5}$ | $\mathbf{52.0}_{10.5}$ | same as |
| | COBias | $6.7_{1.5}$ | $9.2_{0.2}$ | $28.8_{2.0}$ | $\mathbf{14.2}_{1.8}$ | $3.6_{4.3}$ | $25.0_{9.7}$ | full |
| Full | Acc | $\mathbf{87.9}_{0.7}$ | $\mathbf{93.4}_{0.6}$ | $48.3_{1.9}$ | $77.1_{2.0}$ | $74.3_{0.8}$ | $40.4_{6.0}$ | $\mathbf{63.1}_{14.0}$ |
| | COBias | $6.3_{0.6}$ | $\mathbf{7.7}_{0.6}$ | $\mathbf{18.7}_{10.1}$ | $14.2_{1.3}$ | $4.3_{3.3}$ | $\mathbf{7.5}_{3.2}$ | $41.1_{29.6}$ |

Table 3: Comparisons of DNIP results on varying optimization set sizes; min. optimization set size is 10 for all datasets except for DBpedia, which we use 15 to cover its 14 classes. For full optimization set sizes, AGNews: 9,500, DBpedia: 9,500, SST-5: 8,116, TREC: 5,179, RTE: 2,365, DDI: 9,500, PubMedQA: 950.

settings, further showcasing the effectiveness of our approach. For example, the relative average COBias reduction (comparing DNIP to ICL) is 65% and 73% for 1-shot and 5-shot cases respectively, and the relative average accuracy increase is 16% and 19% for 1-shot and 5-shot cases respectively, demonstrating DNIP's capabilities with more shots. In addition, prompt design does contribute to different output probabilities, which could be helpful to provide different starting points for DNIP, so DNIP could optimize for better solutions. However, prompt engineering alone may not be most effective to solve the COBias issue. For example, compared to 1-shot prompting (Table 2), increasing the number of shots does not significantly further reduce COBias or boost accuracy; using more diverse demonstrations (k-shot) may help with COBias, but does not gain much higher accuracy than 1-shot. These results show that more sophisticated ICL settings may only help with the COBias issue to a limited extent, highlighting the necessity of a rigorous COBias reduction method, DNIP.

