# OpenReview forum: "COBias and Debias: Minimizing Language Model Pairwise Accuracy Bias via Nonlinear Integer Programming"
_ICLR.cc/2025/Conference — Submitted to ICLR 2025_

### Official Review · Reviewer_f6uA · 2024-10-27

**Soundness:** 3
**Presentation:** 2
**Contribution:** 2
**Rating:** 5
**Confidence:** 3

**Summary:**

The paper addresses the issue of per-class prediction accuracy imbalance in LLMs by introducing the concept of COBias. COBias measures the difference in accuracy between a class and its "odd" class, which holds the majority of wrong predictions for the former. The authors propose a novel method called Debiasing as Nonlinear Integer Programming (DNIP) to adjust per-class probabilities to reduce COBias while maintaining or improving overall accuracy. The paper reports significant reductions in COBias and improvements in accuracy across three different LLMs on seven NLP classification tasks.

**Strengths:**

1. The research problem is novel and the DNIP method is innovative, leveraging combinatorial optimization to address the debiasing problem directly at the output level.
2. The paper provides extensive experimental results demonstrating the effectiveness of DNIP in reducing COBias and improving accuracy across a variety of LLMs and NLP tasks.

**Weaknesses:**

1. As the metric COBias is also newly proposed in this paper, I think authors should design more baseline methods to validate the effectiveness of the proposed method DNIP. E.g., adapting other debiasing methods.
2. Clarity needs further improvements, for example, font size for labels in Figure 7 is too small.

**Questions:**

1. How does the change in β, τ, K parameters affect the performance of DNIP?
2. The performance of DNIP seems to depend heavily on the number and quality of optimization examples. So if there's limited data or noisy labels, what can we do to better utilize DNIP?

---

> ### Author Response · Authors · 2024-11-20
> **Response to Reviewer f6uA**
>
> Dear Reviewer f6uA,
>
> We sincerely appreciate your precious time and constructive suggestions, and your recognition of the innovation of our approach and the effectiveness of DNIP in reducing COBias and improving accuracy across a variety of LLMs and NLP tasks.
>
> In the following, we would like to answer your concerns separately.
>
> ---
> **Q1**: More baseline debiasing methods.
>
> **R1**: Thank you for the suggestion. Broadly, our work aligns with the existing literature that performs output corrections based on prompted output probabilities, and a major part of existing works is calibration methods. However, to the best of our knowledge, our work significantly differs from calibration methods in debiasing objectives.
>
> The key difference is that calibration techniques aim to correct a model’s imbalanced class prediction to improve overall accuracy, while we abstract the problem as imbalanced class accuracy and correcting it will both improve overall accuracy and reduce the class accuracy imbalance.
>
> In more details, calibration techniques measure the model's bias towards each answer, and directly apply the bias measurements to adjust the probabilities (e.g., via an affine transformation), overlooking the influences of one class on another. Instead, our approach innovatively models the pairwise class accuracy differences, i.e., COBias. Therefore, this presented a challenge for us: the calibration techniques may not explicitly integrate COBias into their methods.
>
> Given these differences in objectives, it is hard to do apples-to-apples comparisons, and it might be best to focus on overall accuracy when comparing calibration methods with DNIP. Weighing all of these, we only compared with a single state-of-the-art calibration method, Batch Calibration (BC), where DNIP outperformed its accuracy on 6 out of 7 benchmark datasets, as shown in Figure 7. (Our COBias was much lower than BC.) Finally, we would like to highlight the lack of in-depth analysis of COBias in the context of ICL, which could stimulate more future work.
>
>
> ---
> **Q2**: How does the change in $\beta, \tau$, K parameters affect the performance of DNIP?
>
>
> **R2**: Thank you for the thoughtful question. We answer for each parameter:
> * $\beta$: It adjusts the COBias term in Equation 4. Increasing it will help achieve more COBias reduction. However, overall accuracy may drop if we only increase it.
> * $\tau$: It adjusts the PMI term in Equation 4. Increasing it will help enhance accuracy. However, COBias may increase if we only increase it.
> * K: the weight scale. We need to point out that **COBias reduction and accuracy enhancement does not scale linearly with a larger weight scale**. In fact, for most of the datasets experimented in this paper, a weight scale of 30 is good enough.
>
> Moreover, there is a dedicated section, Section 4.4, on analyzing how different parameter combinations (objective ablations) affect COBias and accuracy changes, including special settings like $\beta=0$ or $\tau=0$. In short, a good parameter combination selection hinges on one’s goals - whether focusing more on the accuracy aspect, or the COBias aspect.
>
> ---
> **Q3**: If there's limited data or noisy labels, what can we do to better utilize DNIP?
>
>
> **R3**: Thank you for the insightful question. We would like to answer from the following points.
>
> * Limited data: DNIP can achieve great improvements with relatively small optimization set sizes. As demonstrated in Section 4.5 and Table 2 of appendix D, both test accuracy and COBias are improved with only 10 optimization instances; moreover, DNIP has emerged greater COBias reduction abilities with thousands of optimization instances.
>
> * Corrupted data/noisy labels: this is a valuable, practical question that may occur in many downstream tasks. To our understanding, there are methods that are primarily designed for either noisy labels or corrupted samples, e.g., data augmentation, but they lack consideration for capturing the pervasive LLM issue, COBias. This leads to a promising combined method leveraging the best of both worlds. For a simplest combination method, these methods could provide a better starting point and boost the initial solution for DNIP - DNIP could obtain better solutions from a less hassled starting point.
>
> ---
> We will also enlarge Figure 7 to enhance readability. Thanks again for your constructive comments and your recognition of our efforts. We hope the response can clear your concerns.
>
> Best regards,
>
> Authors

---

> > ### Comment · Reviewer_f6uA · 2024-11-22
> > **response to rebuttal**
> >
> > Thank you for your response! After reading your reply, I understand why you do not include previous baselines for comparison.
> >
> > However, in this case, the motivation behind the debias objective of this paper needs to be further discussed. I see that the authors have added an introduction to this objective in the revision, but my primary concern remains the motivation. Why is a more balanced accuracy important? In fact, previous work that "aims to correct a model’s imbalanced class prediction to improve overall accuracy" also uses a validation set with (nearly) equal numbers of samples for each subtask/class. In this context, the overall accuracy itself is already a balanced objective across different classes. This might imply that: 1) The problem addressed in this paper is not as significant; 2) Previous debias methods might also help achieve the objective of this paper (so I still believe it could be worth considering them as baselines)?

---

> > > ### Author Response · Authors · 2024-11-22
> > > **Response to follow-up conerns**
> > >
> > > Thank you for your reply. To address the follow-up concerns:
> > >
> > > 1. Adding more experiments does not strengthen or diminish the novelty of COBias. To our best knowledge, quantifying the pervasive LLM issue of over-predict certain classes while exhibiting diminished performance on others (by Reviewer zeo7) as COBias is novel. In addition, to our best knowledge, no previous methods explicitly target COBias. Therefore, no apples-to-apples comparisons can be made between our approach and previous approaches.
> > >
> > > 2. Why a more balanced per-class accuracy is important: in everyday scenarios, users would expect the same accuracy of what they ask. Suppose a model’s trying to do classifications. If it is good at identifying some classes while having significantly lower prediction accuracy in other classes, some users will have low satisfaction, calling for the need to address the imbalanced per-class accuracy issue as quantified by COBias.
> > >
> > > 3. The focus in DNIP is to balance per-class accuracies while maintaining overall accuracy (and our experimental results show great improvements in both COBias and overall accuracy). No previous methods explicitly target our multiple objectives. Moreover, we hope to spur more future works that consider COBias as a goal in their objectives, and evaluate it.
> > >
> > > Once again, thank you for your time and consideration.
> > >
> > > Best regards,
> > >
> > > Authors

---

> > > > ### Comment · Reviewer_f6uA · 2024-11-27
> > > >
> > > > Thank you for your efforts to address my concerns. I'm still not very convinced by the response, regarding the importance of this research problem. I will maintain my score.

---

> > > > > ### Comment · Reviewer_f6uA · 2024-11-28
> > > > > **to general reponse**
> > > > >
> > > > > Thank you for your response in general response! Yes, what we are discussing is not related to gender bias. However, I believe this issue is quite similar to, and possibly even encompasses, the previous problems of group robustness and worst-group (class) evaluation [1][2]. Given that there are already many solutions to these previous issues, I think it is fundamental to discuss these approaches or consider the possibility of adapting them.
> > > > >
> > > > > [1] Sagawa, S., Koh, P. W., Hashimoto, T. B., et al. (2019). Distributionally robust neural networks for group shifts: On the importance of regularization for worst-case generalization. arXiv preprint arXiv:1911.08731.
> > > > > [2] Liu, E. Z., Haghgoo, B., Chen, A. S., et al. (2021). Just train twice: Improving group robustness without training group information. In International Conference on Machine Learning. PMLR, 6781-6792.

---

> > > > > > ### Author Response · Authors · 2024-11-29
> > > > > > **Thank you for the papers**
> > > > > >
> > > > > > Thank you for the papers you recommended!
> > > > > >
> > > > > > These papers focus on robustness on subgroups of input data in the presence of spurious correlation cues (e.g., grouping together training sentences from the NLI task which do not contain negation words and labeled with “contradictory”, and improve test accuracy of this worst-group), while our work targets class-wise accuracy differences at the output level, no matter if an input sentence contains spurious attributes. Therefore, we have a direct difference in focus.
> > > > > >
> > > > > > On the other hand, paper [1] provides interesting analyses on spurious correlations, especially in the case of NLI, and we will discuss how it could help with finding the causes of class accuracy imbalances on the benchmark datasets studied in our paper. Paper [2]’s upweighting method (JTT) is interesting and not limited to spurious correlations. Therefore, we will think about it and discuss how to combine the method with COBias related objectives.
> > > > > >
> > > > > >
> > > > > >
> > > > > > [1] Sagawa, S., Koh, P. W., Hashimoto, T. B., et al. (2019). Distributionally robust neural networks for group shifts: On the importance of regularization for worst-case generalization. arXiv preprint arXiv:1911.08731.
> > > > > > [2] Liu, E. Z., Haghgoo, B., Chen, A. S., et al. (2021). Just train twice: Improving group robustness without training group information. In International Conference on Machine Learning. PMLR, 6781-6792.

---

### Official Review · Reviewer_zeo7 · 2024-11-02

**Soundness:** 3
**Presentation:** 3
**Contribution:** 3
**Rating:** 6
**Confidence:** 4

**Summary:**

This paper investigates the phenomenon of Contextual Oddity Bias (COBias), wherein large language models (LLMs) exhibit inconsistent accuracy across distinct classes within classification tasks. Despite attaining high overall accuracy in few-shot in-context learning (ICL) settings, LLMs frequently demonstrate a propensity to over-predict certain classes while exhibiting diminished performance on others. This work formally introduces COBias as a quantifiable metric to assess this imbalance, revealing its prevalence even among sophisticated LLMs with diverse architectures and scales.


To mitigate this issue, the authors propose methodology termed Debiasing as Nonlinear Integer Programming (DNIP). DNIP employs a combinatorial optimization approach to refine per-class probabilities through the introduction of corrective weights, explicitly targeting the reduction of COBias while concurrently enhancing overall accuracy.

**Strengths:**

In this paper the authors introduced COBias as a new metric to explicitly quantify the pairwise class accuracy differences that characterize this pervasive LLM issue. By focusing on these pairwise relationships, COBias offers a more nuanced understanding of the accuracy imbalance compared to simply observing disparities in individual class accuracies.

The development of DNIP, a debiasing method based on nonlinear integer programming, represents a creative and innovative solution. Unlike conventional post-hoc correction techniques commonly employed in traditional machine learning, DNIP is specifically tailored to address the unique sources of bias inherent in LLMs, particularly those stemming from in-context learning (ICL) and pre-training data.

The paper shows thorough evaluation. The authors conduct a comprehensive evaluation of DNIP across three different LLMs (GPT-2-XL, Llama-2-7B, and Llama-2-13B), spanning a diverse set of seven NLP tasks, both general and biomedical. This breadth of evaluation demonstrates the generalizability of the proposed approach and its potential applicability across various domains

**Weaknesses:**

The computational cost of DNIP could pose challenges when dealing with tasks involving a large number of classes or when requiring a very fine-grained weight scale. In such scenarios, the optimization process might become prohibitively time-consuming.

The need for powerful hardware to run DNIP efficiently could limit its accessibility for researchers or practitioners with limited computational resources.

Though the authors demonstrate that DNIP can achieve significant improvements even with relatively small optimization set sizes, which suggests that it might be possible to reduce the computational burden by carefully selecting a subset of training data for the optimization process. They may provide some recipe in selecting those subset of training data.

While the paper discusses related work on LLM biases, the direct empirical comparison to other debiasing techniques is somewhat limited.
A more comprehensive comparison with existing calibration techniques, particularly those designed for ICL, would strengthen the paper's claims.

**Minor**
The proposed technique is limited to open-source LLMs due to the unavailability of full probability distributions from closed LLMs

**Questions:**

All the analysis are on 1-shot in-context learning. It would be good if the author can explain why 1-shot in-context leaning is the suitable set up for this problem. They may also consider providing a many-shot (5-shot) analysis, to show whether the proposed method scales well.

Can you share your thoughts of combining DNIP with other debiasing approaches, such as those focusing on prompt engineering or data augmentation, to achieve even greater reductions in COBias and improvements in accuracy.

---

> ### Author Response · Authors · 2024-11-20
> **Response to Reviewer zeo7 (Part 1/2)**
>
> Dear Reviewer zeo7,
>
> We sincerely appreciate your precious time and constructive suggestions. We are greatly encouraged by your high recognition of the innovation of the DNIP approach, the nuanced understanding on accuracy imbalances offered by the COBias metric, and potential applicability across various domains.
>
> In the following, we would like to address the expressed concerns.
>
> ---
> **Q1**: The computational cost of DNIP could pose challenges.
>
> **R1**: Thank you for the insightful feedback. We would like to resolve the concern from your mentioned aspects:
>
> * Regarding challenges when dealing with tasks involving a large number of classes, or when requiring a very fine-grained weight scale:
>     * First, we clarify that **COBias reduction and accuracy enhancement does not scale linearly with a larger weight scale**. In fact, for most of the datasets experimented in this paper, a weight scale of 30 is good enough.
>      * Secondly, we need to mention that, on all the reported datasets, **the computational time is in the scale of minutes**, ranging from a few minutes to dozens of minutes, even for tasks that involve more than 10 classes.
>      * For example, on the 14-class DBpedia task, a weight scale of 90 was experimented but not adopted (when evaluated on the dev set). In addition, when optimizing the full set (9,500 instances) with the 90 scale, the computational time was around 35 minutes. Although a more fine-grained scale will increase number of iterations for the inner loop in Algorithm 1, and an extremely large number of classes could result in a longer computational time, the time elapsed in optimization should be understood as much less than the time for running LLM inferences (that can be up to hours), and it is negligible compared to LLM pre-training time (that can be up to weeks).
>
> * Regarding challenges in the need for powerful hardware to run DNIP efficiently: to run DNIP, we just need CPUs, which can be accessed on both servers and personal laptops. Although we use Python to perform the experiments, the DNIP algorithm could be readily rewritten in C or other compatible programming languages, allowing fast computation on an individual’s computer.
>
> ---
> **Q2**: DNIP can achieve significant improvements with relatively small optimization set sizes, which might help reduce the computational costs. The authors may provide some recipe in selecting those subset of training data.
>
> **R2**: Thank you for the great suggestion. Optimization set size is a factor that affects computational time. Smaller optimization set sizes can indeed reduce computational times to seconds. For example, on DBpedia, the computational time is 3 seconds for an optimization set size of 15, and several minutes for an optimization set size of 1,000. For subset selection, we try to use a balanced sample size for each class to ensure best optimization results.
>
>
> ---
> **Q3**: The authors may also consider providing a many-shot (5-shot) analysis, to show whether the proposed method scales well.
>
> **R3**: Thank you for your very constructive suggestion. Please find the additional 5-shot analysis in the table below. DNIP significantly reduces COBias and improves accuracy in the 5-shot setting, further showcasing the effectiveness of our approach.
>
> |Prompting Method  |Metric  | AGNews | DBpedia | SST-5 | TREC | RTE | DDI | PubMedQA | Avg. |
> |---|---|:---:|---|---|:---:|---|:---:|---|---|
> | 1-shot (random) ICL | Acc | $79.9_{7.0}$ | $88.6_{1.7}$ | $44.9_{4.3}$ | $68.5_{10.8}$ | $71.5_{2.2}$ | $7.2_{0.9}$ | $55.1_{2.9}$ | 59.4 |
> |  | COBias | $28.3_{16.1}$ | $16.2_{3.7}$ | $53.1_{5.0}$ | $35.9_{6.5}$ | $43.4_{7.0}$ | $45.6_{5.9}$ | $61.2_{1.9}$ | 40.5 |
> | 1-shot (random) DNIP | Acc | $\boldsymbol{87.9_{0.7}}$ | $\boldsymbol{93.4_{0.6}}$ | $\boldsymbol{48.3_{1.9}}$ | $\boldsymbol{77.1_{2.0}}$ | $\boldsymbol{74.3_{0.8}}$ | $\boldsymbol{40.4_{6.0}}$ | $\boldsymbol{63.1_{14.0}}$ | 69.2 |
> |  | COBias | $\boldsymbol{6.3_{0.6}}$ | $\boldsymbol{7.7_{0.6}}$ | $\boldsymbol{18.7_{10.1}}$ | $\boldsymbol{14.2_{1.3}}$ | $\boldsymbol{4.3_{3.3}}$ | $\boldsymbol{7.5_{3.2}}$ | $\boldsymbol{41.1_{29.6}}$ | 14.3 |
> | 5-shot (random) ICL | Acc | $82.5_{2.0}$ | $93.6_{1.3}$ | $45.8_{6.0}$ | $58.7_{23.3}$ | $61.9_{16.9}$ | $34.1_{42.1}$ | $44.7_{5.7}$ | 60.2 |
> |  | COBias | $16.5_{6.0}$ | $9.0_{2.0}$ | $48.0_{15.4}$ | $35.4_{13.7}$ | $61.3_{40.4}$ | $44.4_{5.4}$ | $52.2_{19.3}$ | 38.1 |
> | 5-shot (random) DNIP | Acc | $\boldsymbol{88.5_{0.6}}$ | $\boldsymbol{95.8_{0.7}}$ | $\boldsymbol{52.9_{2.5}}$ | $\boldsymbol{76.6_{8.7}}$ | $\boldsymbol{75.8_{4.5}}$ | $\boldsymbol{54.1_{5.1}}$ | $\boldsymbol{59.9_{1.4}}$ | 71.9 |
> |  | COBias | $\boldsymbol{7.0_{0.9}}$ | $\boldsymbol{5.7_{1.1}}$ | $\boldsymbol{15.8_{12.6}}$ | $\boldsymbol{14.4_{7.5}}$ | $\boldsymbol{3.1_{1.9}}$ | $\boldsymbol{16.5_{7.6}}$ | $\boldsymbol{9.6_{4.6}}$ | 10.3 |
>
> Table A-zeo7. Additional results using 5-shot prompting. Average score with standard deviation over three runs are reported.

---

> > ### Author Response · Authors · 2024-11-20
> > **Response to Reviewer zeo7 (Part 2/2)**
> >
> > **Q4**: A more comprehensive comparison with existing calibration techniques, particularly those designed for ICL, would strengthen the paper's claims.
> >
> > **R4**: Thank you for the feedback. This was a dilemma for us: the calibration techniques may not explicitly model COBias in their methods. Broadly, our paper aligns with the literature that performs output corrections based on prompted output probabilities. However, to the best of our knowledge, our work significantly differs from calibration methods in debiasing objectives.
> >
> > The key difference is that calibration techniques aim to correct a model’s imbalanced class prediction to improve overall accuracy, while we abstract the problem as imbalanced class accuracy and correcting it will both improve overall accuracy and reduce the class accuracy imbalance.
> >
> > In more details, calibration techniques measure the model's bias towards each answer, and directly apply the measurements to adjust the probabilities (e.g., via an affine transformation), **overlooking the influences of one class on another**. As your review pointed out, our approach innovatively models the pairwise class accuracy differences, i.e., COBias.
> >
> > Given these differences in objectives, it is hard to do apples-to-apples comparisons, and it might be best to mainly compare overall accuracy between calibration methods and DNIP. Weighing all of these, we only compared with a single state-of-the-art calibration method, Batch Calibration (BC), where DNIP outperformed its accuracy on 6 out of 7 benchmark datasets, as shown in Figure 7. (Our COBias was much lower than BC.) Finally, we would like to highlight the lack of in-depth analysis of COBias in the context of ICL, which could stimulate more future work.
> >
> > ---
> > **Q5**: Can you share your thoughts of combining DNIP with other debiasing approaches, such as those focusing on prompt engineering or data augmentation, to achieve even greater reductions in COBias and improvements in accuracy?
> >
> > **R5**: Thank you for the thoughtful question. We would like to answer it from the nature of DNIP and the mentioned approaches. Prompt engineering or data augmentation can be viewed as pre-hoc correction techniques, whereas DNIP is a post-hoc correction technique, and what matters is a starting point for optimization. As long as there is a need to balance per-class performances, DNIP can further optimize on per-class probabilities obtained by models that apply prompt engineering or data augmentation techniques. Furthermore, calibration methods may also provide a good starting point for DNIP. In addition, data augmentation may help with building effective optimization subsets, which will be left as future work.
> >
> > ---
> > Thanks again for your constructive comments and your recognition of our efforts. We hope the response can resolve your concerns.
> >
> >
> > Best regards,
> >
> > Authors

---

> > > ### Author Response · Authors · 2024-11-24
> > > **A kind reminder for feedback**
> > >
> > > Dear Reviewer zeo7,
> > >
> > > Thanks again for reviewing our work. Please let us know if our response has adequately addressed your questions and concerns.
> > >
> > > Sincerely,
> > >
> > > Authors

---

> > > > ### Author Response · Authors · 2024-12-04
> > > >
> > > > Dear Reviewer zeo7,
> > > >
> > > > We would appreciate a re-evaluation if you find our rebuttal and the revised manuscript strengthens the paper. We have always appreciated your kind words and insightful questions.
> > > >
> > > >
> > > > Sincerely,
> > > >
> > > > Authors

---

### Official Review · Reviewer_Ntx9 · 2024-11-04

**Soundness:** 3
**Presentation:** 1
**Contribution:** 3
**Rating:** 3
**Confidence:** 3

**Summary:**

Within few-shot in-context learning (ICL), LLMs can achieve great performance for classification tasks. However the raw accuracy can hide prohibitive differences in per-class accuracies. In this paper, this phenomenon is defined as Contextual Oddity Bias (COBias). COBias is a proposed measure of the bias in prediction. Then the authors explore the use of Nonlinear Integer Programming for debiasing existing LLM. The results are reported on seven NLP classification tasks with success.

**Strengths:**

The paper consider the per-class imbalance in performance for in-context learning classification, as a  bias embedded by the model. This is a nice idea. Then the proposition to debias the model with nonlinear integer programming is also a promising contribution. The experimental results show great improvement in both accuracy and debiasing.

**Weaknesses:**

There are two main weaknesses.  The first one is about clarity. The paper is sometimes  difficult to read and sentences seem really obscure. Just to take the abstract as example, the first two sentences are really confusing. Maybe it is a problem of order in the ideas. These two first sentences could be easier to understand later in the paper, with more context,  but as a starting point it is really unclear. I will add more examples in the questions part.
Consider: "For language model classification, would you prefer having only one workable class or having every class working? The latter makes more practical uses."  You want to classify language models ? or you want to perform text classification with LLMs ? What do you mean by "workable" here ? and so on.

We have the same issue with the  first two sentences of the introduction: "We rethink language models’ imbalance in per-class prediction accuracy and reconceptualize it as the Contextual Oddity Bias (COBias). In a nutshell, COBias refers to the difference in accuracy
by a class A compared to its “odd” class, which holds the majority wrong predictions of class A."  To start a paper, more context would be appreciated.

My second and  scientific concern is about the equation 1. COBias is presented as an attempt to detect when a class is not predicted while it is the good one, and in favor of another one.  It is not completely clear why the equation 1 measures this kind of bias. The definition of the odd class is very interesting: "an odd class to be a class at inference time where predictions of another class are often biased towards it. An odd class is relative to the other class.". However equation 1 does not quantify exactly that, since it relies only on per-class accuracies.
Moreover, the absolute value ignores the fact that one class is better classified than the other.

At the end I think that this paper contains nice contributions that deserve a complete rewrite of the paper.

**Questions:**

You could open the paper by " For large language models (LLMs), the fact that
they achieve remarkable test accuracy over all class labels by few-shot in-context learning (ICL)
(Brown et al., 2020) can obscure a large difference in individual class accuracies."

The name "liking" is usually followed by "for"

l 161 : it times a correction weight ...  I don't understand the verb times here ?


Could you define the pointwise mutual information term more precisely ? The footnote helps, but it is not sufficient to understand the motivation of this term, its importance and the sensitivity to the smoothing factor.
The paragraph starting at line 194 remains obscure to me while it is an important aspect of the contributions.

The choice of simulated annealing is not straightforward. I would think that Branch-and-Bound algorithm are more efficient for this kind of optimization program. However beyond efficiency, maybe there is another motivation in this choice ?

The definition of the odd class looks like an adversarial class maybe you could comment on this since there is a large body of work on that topic.

---

> ### Author Response · Authors · 2024-11-20
> **Response to Reviewer Ntx9**
>
> Dear Reviewer Ntx9,
>
> We truly appreciate your precious time, constructive suggestions, and your recognition that this paper contains nice contributions.
>
> ---
> **Q1**: Clarity about writing.
>
> **R1**: Thank you for your feedback. We are surprised and sorry for making you feel confused. To address some of the probably most confusing parts:
>
> * The first sentence in the abstract was a rhetorical question referring to the per-class accuracy imbalance issue when performing classification tasks with language models. We will change “For language model classification” to “When performing classification tasks with language models”.
>
> * Thanks for the suggestion about how to open the paper. We will follow your advice to move “For large language models (LLMs), the fact that they achieve remarkable test accuracy over all class labels by few-shot in-context learning (ICL) can obscure a large difference in individual class accuracies.” to the beginning.
>
> * We will correct grammatical errors in our writings following your suggestions.
>
>
> ---
> **Q2**: Clarity about Equation 1 (COBias**single**) and the exact definition of COBias.
>
> **R2**: What motivates Equation 1 (COBias**single**) is a subtle, often overlooked observation that a most frequently predicted class can hold the majority of wrong predictions of other classes, especially of a least frequently predicted class; we also empirically find a difference in the most and least frequently predicted classes’ accuracies. Therefore, for this single pair of classes, we define the class (most frequently predicted) that the other (least frequently predicted) is most biased towards as an odd class, and their accuracy difference as COBias**single**. Note here, the most frequently predicted class could also be most/secondly most/… biased towards the least frequently predicted class. Hence, the absolute difference is taken to reflect both directions: “A is biased towards B” and “B is biased towards A”.
>
> We later realize that the per-class accuracy differences is *the issue* to solve for enhancing LLMs’ text classification abilities, and we generalize the above measure to every pair of classes, obtaining **the proposed COBias metric, which is Equation 2**. Indeed, the accuracy difference of some pairs in Equation 2 does not reflect a strong biasedness between the two classes. We continue using the name “COBias” just to honor the observation.
>
> Back to your concern of clarity, we would be happy to remove Equation 1 upon request to highlight the more general scope of our insights.
>
>
> ---
> **Q3**: Motivation of the PMI term.
>
> **R3**: Thanks for the question. What motivates the PMI term in Equation 4 is a goal to enforce per-class prediction to be close to its actual class. When it is far away from the actual class, we penalize it.
>
> ---
> **Q4**: The choice of simulated annealing is not straightforward. I would think that Branch-and-Bound algorithm are more efficient for this kind of optimization program. However beyond efficiency, maybe there is another motivation in this choice?
>
> **R4**: Thank you for your question. Our DNIP model is not convex. The branch-and-bound (BnB) algorithm is commonly used for solving linear integer programming problems; it could be difficult to solve our model with BnB. Instead, Simulated Annealing is a metaheuristic that more easily solves our model.
>
>
> ---
> **Q5**: The definition of the odd class looks like an adversarial class maybe you could comment on this.
>
> **R5**: Thanks for the comment. The odd class as defined in COBias**single** is a most frequently predicted class, so it is not an adversarial class per se.
>
> However, this viewpoint is very interesting in that it could go deeper into the root cause of the observed imbalanced per-class accuracies, i.e., “unreasonable” overprediction of some classes. To our understanding, the instances belonging to a downstream task may present some unique tokens/patterns that can "fool" the model to favor a particular class, like an adversarial attack. Which patterns trigger those adversarial behaviors could depend on the LLM and also how you query it (prompt related aspects). This viewpoint could open a body of works that combine adversarial attack techniques with integer programming to further enhance LLM predictions.
>
>
> ---
> Thanks again for your constructive comments and your recognition of our efforts. We hope the response can resolve your concerns.
>
> Best regards,
>
> Authors

---

> > ### Author Response · Authors · 2024-11-24
> > **A kind reminder for feedback**
> >
> > Dear Reviewer Ntx9,
> >
> > Thanks again for reviewing our work. Please let us know if our response has adequately addressed your questions and concerns.
> >
> > Sincerely,
> >
> > Authors

---

> > > ### Author Response · Authors · 2024-11-30
> > >
> > > Dear Reviewer Ntx9,
> > >
> > > We sincerely appreciate your valuable contributions as our reviewer and thank you for your time and insights.
> > >
> > > We kindly request a re-evaluation of our work. We have addressed your concerns with the rebuttal and the revised manuscript, and hope these clarifications will resolve any misunderstandings, particularly regarding the clarity of writing.
> > >
> > > Once again, we extend our gratitude for your hard work in making ICLR a success this year.
> > >
> > > Best regards,
> > >
> > > Authors

---

> > > > ### Author Response · Authors · 2024-12-04
> > > >
> > > > Dear Reviewer Ntx9,
> > > >
> > > > We would appreciate a re-evaluation if you find our rebuttal and the revised manuscript strengthens the paper. We have always appreciated your great questions and suggestions.
> > > >
> > > > Sincerely,
> > > >
> > > > Authors

---

### Official Review · Reviewer_fcyJ · 2024-11-06

**Soundness:** 3
**Presentation:** 3
**Contribution:** 3
**Rating:** 6
**Confidence:** 2

**Summary:**

The paper shows that accuracy metric hides large differences in individual class accuracies. Focusing on the LLM In Context Learning setting,  the paper defines a new metric called COBias to uncover and measure this imbalance in per class prediction accuracy.
Further, the authors propose a technique called DNIP (based on integer programming) to mitigate this bias problem by post-hoc correcting the predicted class probabilities. DNIP greatly reduces the COBias while also leading to large accuracy gains.

**Strengths:**

- The paper studies a really cool topic that is very important but quite under-explored.
- We know that the widely reported accuracy metric probably masks lot of per class / inter-class details. This by itself if not at all surprising, but the authors quantitatively show that this does happen a lot.
- The COBias metric definition makes intuitive sense and is easy to compute.
- Strong empirical results on a variety of benchmarks. What's cool is that the DNIP technique not only reduces COBias but also leads to increase in overall accuracy.
- This contribution could spur lot of follow-up work. Could also encourage researchers to go beyond accuracy when evaluating results in a classification task.

**Weaknesses:**

The ICL setup only uses 1-shot example. Given that the main motivation for the paper is demonstrate and mitigate COBias in the ICL setting, I would have expected more baselines. Eg: How does COBias change as you increase the number of shots? What happens in the case where you ensure that each class is represented by at least one example in the prompt? How does more sophisticated prompt engineering impact COBias and DNIP?
Adding these baselines and investigation would improve the paper greatly.

Minor point: In terms of metrics, all the focus is on COBias and Accuracy. But it would have been helpful to at least contrast with some other metrics like macro F1 and micro F1 - do they also uncover issues masked by accuracy?

**Questions:**

Is COBias also an issue when LLMs are fine-tuned?

---

> ### Author Response · Authors · 2024-11-20
> **Response to Reviewer fcyJ (Part 1/2)**
>
> Dear Reviewer fcyJ,
>
> We truly appreciate your precious time and constructive suggestions. We are deeply encouraged by your high recognition of the importance of the COBias metric, strong empirical results led by the DNIP method, and a potential of spurring follow-up works and going beyond accuracy for classification evaluations.
>
> In the following, we would like to answer your concerns.
>
> ---
> **Q1**: The ICL setup only uses 1-shot example. How does COBias change as you increase the number of shots? What happens when you ensure that each class is represented by at least one example in the prompt? And what about more sophisticated prompt engineering?
>
> **R1**: Thank you for the very insightful questions. To better address them, we performed additional experiments.
>
> We prompted Llama-2-13B (using a single 80G A100 GPU) in two additional settings: **5-shot (random)**, where the number of shots is increased to 5, and each demonstration is randomly selected as in the 1-shot case; **k-shot (1 demonstration from each class)**, where k is the number of classes and each class is represented by a demonstration example in the prompt.
>
> Table A presents results on 7 benchmark datasets in both settings. The main findings are:
>
> * DNIP significantly reduces COBias and improves accuracy in both 5-shot and k-shot settings, further showcasing the effectiveness of our approach. For example, the relative average COBias reduction (ICL $\rightarrow$ DNIP) is 65% and 73% for 1-shot and 5-shot cases respectively, and the relative average accuracy increase is 16% and 19% for 1-shot and 5-shot cases respectively, demonstrating DNIP’s capabilities with more shots.
>
> * Prompt design does contribute to different output probabilities, which could be helpful to provide different starting points for DNIP, so DNIP could optimize for better solutions. However, prompt engineering alone may not be most effective to solve the COBias issue.
>
> 	* Compared to 1-shot prompting, increasing the number of shots does not significantly further reduce COBias or boost accuracy; using more diverse demonstrations (k-shot) may help with COBias, but does not gain much higher accuracy than 1-shot.
> 	* These results show that the two settings can only help with the COBias issue to a limited extent, highlighting the necessity of a rigorous COBias reduction method - DNIP.
>
> |Prompting Method  |Metric  | AGNews | DBpedia | SST-5 | TREC | RTE | DDI | PubMedQA | Avg. |
> |---|---|:---:|---|---|:---:|---|:---:|---|---|
> | 1-shot (random) ICL | Acc | $79.9_{7.0}$ | $88.6_{1.7}$ | $44.9_{4.3}$ | $68.5_{10.8}$ | $71.5_{2.2}$ | $7.2_{0.9}$ | $55.1_{2.9}$ | 59.4 |
> |  | COBias | $28.3_{16.1}$ | $16.2_{3.7}$ | $53.1_{5.0}$ | $35.9_{6.5}$ | $43.4_{7.0}$ | $45.6_{5.9}$ | $61.2_{1.9}$ | 40.5 |
> | 1-shot (random) DNIP | Acc | $\boldsymbol{87.9_{0.7}}$ | $\boldsymbol{93.4_{0.6}}$ | $\boldsymbol{48.3_{1.9}}$ | $\boldsymbol{77.1_{2.0}}$ | $\boldsymbol{74.3_{0.8}}$ | $\boldsymbol{40.4_{6.0}}$ | $\boldsymbol{63.1_{14.0}}$ | 69.2 |
> |  | COBias | $\boldsymbol{6.3_{0.6}}$ | $\boldsymbol{7.7_{0.6}}$ | $\boldsymbol{18.7_{10.1}}$ | $\boldsymbol{14.2_{1.3}}$ | $\boldsymbol{4.3_{3.3}}$ | $\boldsymbol{7.5_{3.2}}$ | $\boldsymbol{41.1_{29.6}}$ | 14.3 |
> | 5-shot (random) ICL | Acc | $82.5_{2.0}$ | $93.6_{1.3}$ | $45.8_{6.0}$ | $58.7_{23.3}$ | $61.9_{16.9}$ | $34.1_{42.1}$ | $44.7_{5.7}$ | 60.2 |
> |  | COBias | $16.5_{6.0}$ | $9.0_{2.0}$ | $48.0_{15.4}$ | $35.4_{13.7}$ | $61.3_{40.4}$ | $44.4_{5.4}$ | $52.2_{19.3}$ | 38.1 |
> | 5-shot (random) DNIP | Acc | $\boldsymbol{88.5_{0.6}}$ | $\boldsymbol{95.8_{0.7}}$ | $\boldsymbol{52.9_{2.5}}$ | $\boldsymbol{76.6_{8.7}}$ | $\boldsymbol{75.8_{4.5}}$ | $\boldsymbol{54.1_{5.1}}$ | $\boldsymbol{59.9_{1.4}}$ | 71.9 |
> |  | COBias | $\boldsymbol{7.0_{0.9}}$ | $\boldsymbol{5.7_{1.1}}$ | $\boldsymbol{15.8_{12.6}}$ | $\boldsymbol{14.4_{7.5}}$ | $\boldsymbol{3.1_{1.9}}$ | $\boldsymbol{16.5_{7.6}}$ | $\boldsymbol{9.6_{4.6}}$ | 10.3 |
> | k-shot (1 demon.  from each class) ICL | Acc | $83.5_{1.5}$ | $95.2_{1.2}$ | $50.3_{2.3}$ | $67.0_{12.7}$ | $75.0_{0.8}$ | $9.7_{1.0}$ | $52.3_{5.3}$ | 61.9 |
> |  | COBias | $14.9_{5.1}$ | $7.0_{2.2}$ | $36.3_{7.2}$ | $38.2_{5.1}$ | $22.5_{13.2}$ | $39.7_{3.5}$ | $20.9_{4.2}$ | 25.6 |
> | k-shot (1 demon.  from each class) DNIP | Acc | $\boldsymbol{88.7_{0.5}}$ | $\boldsymbol{96.6_{0.5}}$ | $\boldsymbol{51.3_{1.0}}$ | $\boldsymbol{82.7_{1.4}}$ | $\boldsymbol{76.7_{3.7}}$ | $\boldsymbol{43.6_{6.1}}$ | $\boldsymbol{55.3_{2.6}}$ | 70.7 |
> |  | COBias | $\boldsymbol{7.3_{0.5}}$ | $\boldsymbol{4.3_{0.7}}$ | $\boldsymbol{2.8_{1.5}}$ | $\boldsymbol{12.1_{5.5}}$ | $\boldsymbol{5.0_{3.3}}$ | $\boldsymbol{12.5_{4.3}}$ | $\boldsymbol{8.7_{1.2}}$ | 7.5 |
>
> Table A-fcyJ. Comparison on different shots. Average score with standard deviation over three runs are reported.

---

> > ### Author Response · Authors · 2024-11-20
> > **Response to Reviewer fcyJ (Part 2/2)**
> >
> > **Q2**: Can metrics like F1 scores also uncover issues masked by accuracy?
> >
> > **R2**: Thank you for the thoughtful question. F1 score seems to contain more information than just accuracy, as it provides a balanced blend of precision and recall. However, it does not uncover more information for the imbalanced per-class accuracy issue. This is seen from the following two aspects.
> >
> > * Per-class F1 does not express more useful information than per-class accuracy. By knowing the recall/acc of a class, we also gain information about the precision of another class. Thus, per-class precision, or per-class F1, might not be necessary. To see it, we illustrate with Llama-2-13B’s predictions on the AGNews test set (using random seed 0 for 1-shot prompting). All metrics are computed by Python scikit-learn. Note that by scikit-learn, per-class recall is the same as per-class accuracy. On AGNews, class 2 has a relatively low precision of 0.55, which is co-reflected by class 3’s relatively low recall/acc of 0.19. The reason is, out of 1240 class 3 instances, a majority of 822 instances were wrongly predicted as class 2, leading to low precision of class 2 and low recall/acc of class 3. Therefore, per-class accuracies are essential indicators for per-class performances.
> >
> > | Class | Precision | Recall/Accuracy | F1-score | Support |
> > |---|---|---|---|---|
> > | 0 | 0.85 | 0.85 | 0.85 | 1286 |
> > | 1 | 0.93 | 0.98 | 0.95 | 1270 |
> > | 2 | 0.55 | 0.97 | 0.70 | 1204 |
> > | 3 | 0.96 | 0.19 | 0.32 | 1240 |
> > | Accuracy |  |  | 0.75 | 5000 |
> > | Macro Avg. | 0.82 | 0.75 | 0.71 | 5000 |
> >
> >
> > |  |  |  | Pred |  |  |
> > |---|---|---|---|---|---|
> > |  |  | 0 | 1 | 2 | 3 |
> > |  | 0 | 1093 | 64 | 126 | 3 |
> > | **True** | 1 | 9 | 1247 | 14 | 0 |
> > |  | 2 | 25 | 4 | 1167 | 8 |
> > |  | 3 | 156 | 27 | $\boldsymbol{822}$ | 235 |
> >
> >
> > * Overall F1 scores may instead mask the imbalanced per-class accuracy issue. For example, macro-F1, taking the arithmetic mean of all per-class F1 scores, does not show the exact performance gaps between classes. Overall F1, like overall accuracy, may hinder the per-class accuracy imbalances, further suggesting the need of pairwise per-class accuracy difference measurements, i.e., the COBias metric.
> >
> > ---
> > **Q3**: Is COBias also an issue when LLMs are fine-tuned?
> >
> > **R3**: Thank you for the question. Yes, the imbalanced per-class accuracy issue also happens in fine-tuned language models as discussed in earlier works. Paper [1] shows that a prompt-based fine-tuned BERT exhibits a prediction bias towards the majority class in the fine-tuning data. Current LLMs, though being magnitudes of larger than BERT, can hold similar findings when prompt-based learning/fine-tuning techniques are applied.
> >
> > 1. Ruixi Lin and Hwee Tou Ng. “Mind the Biases: Quantifying Cognitive Biases in Language Model Prompting.” Findings of the Association for Computational Linguistics (2023): 5269–5281
> > ---
> > Thanks again for your constructive comments and your recognition of our efforts. We hope the response can address your concerns.
> >
> > Best regards,
> >
> > Authors

---

> > > ### Author Response · Authors · 2024-11-24
> > > **A kind reminder for feedback**
> > >
> > > Dear Reviewer fcyJ,
> > >
> > > Thanks again for reviewing our work. Please let us know if our response has adequately addressed your questions and concerns.
> > >
> > > Sincerely,
> > >
> > > Authors

---

> > > > ### Author Response · Authors · 2024-12-04
> > > >
> > > > Dear Reviewer fcyJ,
> > > >
> > > > We would appreciate a re-evaluation if you find our rebuttal and the revised manuscript strengthens the paper. We have always appreciated your kind words and insightful questions.
> > > >
> > > >
> > > > Sincerely,
> > > >
> > > > Authors

---

### Author Response · Authors · 2024-11-22
**Revised Manuscript Uploaded**

Dear Reviewers and AC,

Thank you for your precious time in handling our submission. We appreciate the professional comments and suggestions from the reviewers, and are encouraged by their recognition of the novelty and contributions of our approach.

We address the concerns in the rebuttal, and have uploaded a revised manuscript, where revisions made are highlighted in fuchsia purple color.

**Revisions made:**

1. Clarity of abstract and introduction (following Reviewer Ntx9’s suggestion)
2. Clarity of COBias definition, the choice of simulated annealing, and grammatical errors  (following Reviewer Ntx9’s suggestion)
3. Added a discussion on computational costs (following Reviewer zeo7’s suggestion)
4. Added results and discussion on more ICL setups (following Reviewer fcyJ and zeo7’s suggestion)
5. Enlarged Figure 7 (following Reviewer f6uA’s suggestion)

Best regards,

Authors

---

### Author Response · Authors · 2024-11-27
**General Response**

**About the importance of this work:**

Mitigating class accuracy imbalance is under-explored, but it is as important as mitigating more well-explored answer biases such as gender bias. It is known that LLMs are infamous for over-predicting certain classes. This unwanted answer bias results in imbalanced class accuracy, which can cause serious damages. For example, a patient queries an LLM about drug-drug interactions, but he/she doesn’t know the model over-predicts “no interaction” and has low accuracy for other classes of interactions. If the patient receives “no interaction” and trusts it, and the true answer is a kind of interaction, then taking the two drugs together can put one’s life at risk in some cases. Therefore, it is high time we took actions to reduce such biases (and it’d be better if we could mitigate these biases without hurting overall accuracy).

What we tackle is such an under-explored LLM issue, and what we propose is a new quantifiable metric of pairwise class accuracy bias (COBias) and a new method based on the metric to tackle the issue directly from LLM outputs. This paper’s idea is not incremental like A+B, where A is an established prior work and B is something new, so there isn’t such baselines for us to compare, and the importance of this research should not be validated according to prior work that did not aim what we aimed. The importance has been validated in this work’s superior improvements over the original undebiased ICL results, and it could also be further verified in the future - as some reviewers suggest, this work could encourage follow-up works.

Thanks to you all!

---

### Meta-Review · Area_Chair_QtPA · 2024-12-22

**Metareview:**

The paper presents an innovative approach to addressing an under-explored issue in LLMs, namely the per-class prediction accuracy imbalance. most of the reviewers recognized the paper's focus on an under-explored yet important topic, the intuitive and easy-to-compute COBias metric, strong empirical results showing DNIP's effectiveness in reducing COBias and increasing overall accuracy.  However, I agree with reviewer Ntx9's comments that this paper should be further improved in the clarification, and author should also *intuitively* explain the  Equation 1. I recommend authors to heavily fix the wiring issues, which will make this manuscript stronger in the next submission cycle.

**Additional Comments On Reviewer Discussion:**

1) Authors conducted additional experiments in 5-shot and k-shot settings, demonstrating DNIP's continued effectiveness.
2) They also explained that F1 scores don't uncover more information on the imbalanced per-class accuracy issue compared to per-class accuracies.
3) The authors also tried to address computational cost concerns by clarifying the relationship between COBias reduction and weight scale, as well as explained the difficulty in direct comparison with calibration methods due to different debiasing objectives and compared with a state-of-the-art calibration method.

---

### Decision · Program_Chairs · 2025-01-22

Reject